# Reconstruction of Par-dependent polarity in apolar cells reveals a dynamic process of cortical polarization

Kalyn Kono[1,2], Shigeki Yoshiura[2†], Ikumi Fujita[2†], Yasushi Okada[3,4,5], Atsunori Shitamukai[2], Tatsuo Shibata[6], Fumio Matsuzaki[1,2]*

[1]Laboratory of Molecular Cell Biology and Development, Department of Animal Development and Physiology, Graduate School of Biostudies, Kyoto University, Kyoto, Japan; [2]Laboratory for Cell Asymmetry, RIKEN Center for Biosystems Dynamics Research, Kobe, Japan; [3]Laboratory for Cell Polarity Regulation, RIKEN Center for Biosystems Dynamics Research, Osaka, Japan; [4]Department of Physics, Graduate School of Science, The University of Tokyo, Tokyo, Japan; [5]Universal Biology Institute, Graduate School of Science, The University of Tokyo, Tokyo, Japan; [6]Laboratory for Physical Biology, RIKEN Center for Biosystems Dynamics Research, Kobe, Japan

**Abstract** Cellular polarization is fundamental for various biological processes. The Par network system is conserved for cellular polarization. Its core complex consists of Par3, Par6, and aPKC. However, the general dynamic processes that occur during polarization are not well understood. Here, we reconstructed Par-dependent polarity using non-polarized *Drosophila* S2 cells expressing all three components endogenously in the cytoplasm. The results indicated that elevated Par3 expression induces cortical localization of the Par-complex at the interphase. Its asymmetric distribution goes through three steps: emergence of cortical dots, development of island-like structures with dynamic amorphous shapes, repeating fusion and fission, and polarized clustering of the islands. Our findings also showed that these islands contain a meshwork of unit-like segments. Furthermore, Par-complex patches resembling Par-islands exist in *Drosophila* mitotic neuroblasts. Thus, this reconstruction system provides an experimental paradigm to study features of the assembly process and structure of Par-dependent cell-autonomous polarity.

DOI: https://doi.org/10.7554/eLife.45559.001

**\*For correspondence:**
fumio.matsuzaki@riken.jp

†These authors contributed equally to this work

**Competing interests:** The authors declare that no competing interests exist.

## Introduction

Polarization is a fundamental cellular property that plays a vital role in various biological processes in multi-cellular as well as single-cell organisms. Par-complex system is a conserved mechanism that regulates cell polarization (*Kemphues et al., 1988*; *Suzuki and Ohno, 2006*; *St Johnston, 2018*). The core Par-complex consists of Par6, Par3, and atypical protein kinase C (aPKC) (*Kemphues et al., 1988*; *Tabuse et al., 1998*). Domain structures of these components and their interactions have been extensively studied (*Lang and Munro, 2017*). Par3 exhibits membrane-binding affinity through its C-terminal domain and the ability to self-oligomerize via its N-terminal CR1 domain, which is essential for its localization and function (*Benton and St Johnston, 2003*; *Mizuno et al., 2003*; *Krahn et al., 2010*; *Harris, 2017*). Structural studies have revealed that the CR1 domain forms helical polymers of 10 nm diameter (*Zhang et al., 2013*). Par6 and aPKC, which form a stable subcomplex, interact with the CR3 and PDZ domains of Par3 (*Izumi et al., 1998*; *Renschler et al., 2018*). Phosphorylation of this domain by aPKC inhibits this interaction (*Morais-de-Sá et al., 2010*; *Soriano et al., 2016*). Thus, Par-complex assembly is a dynamic process. Cdc42 binds to the aPKC-

Par6 subcomplex and anchors it to the cell membrane as a diffusible cortical form (*Joberty et al., 2000*; *Aceto et al., 2006*; *Rodriguez et al., 2017*; *Wang et al., 2017*) On the other hand, Lgl and/ or Par1 kinase act as inhibitory factors against aPKC (*Guo and Kemphues, 1995*; *Betschinger et al., 2003*; *Yamanaka et al., 2003*; *Plant et al., 2003*; *Hurov et al., 2004*), and distribute complementarily to the core Par complex. Interplay between these components results in cytocortical asymmetry (*Doerflinger et al., 2006*; *Sailer et al., 2015*).

Cell polarization involving the Par-complex in situ is linked to various other processes. The Par-complex creates epithelial cell polarity during interphase at the subapical domain (including tight junctions) that is tightly associated with adherens junctions, where Par3 primarily localizes (*Rodriguez-Boulan and Macara, 2014*; *Suzuki and Ohno, 2006*). On the other hand, cell polarization is coupled with mitosis during asymmetric divisions, and autonomously induced or triggered by an external cue, depending on the cell type (*Yamashita et al., 2010*). Because of such association between Par-dependent polarization and other processes, the Par-complex exhibits different behavioral characteristics in an individual context, making it difficult to determine general features of the dynamic process taking place during cell polarization by the Par-complex. To understand the general and dynamic characteristics of the cell polarization process induced by the Par-complex, we attempted to reconstruct Par-complex-dependent cell polarization system in a cell-autonomous manner using non-polar cells. We used *Drosophila* Schneider cells (S2 cells) of mesodermal origin, as host cells for cell-autonomous reconstruction of cell polarity (*Schneider, 1972*). They are neither polarized nor adhere to the substratum and between cells. To date, Baas *et al.* reconstructed epithelial cells that can form epithelial sheets from a cultured cell line bearing a partial epithelial character (*Baas et al., 2004*). Johnston *et al.* expressed a fusion protein of a cell adhesion molecule and aPKC in S2 cells, allowing a pair of transfected S2 cells to adhere each other via this fusion protein (*Johnston et al., 2009*). This resulted in the restricted cortical distribution of aPKC in the non-cell autonomous manner. However, there is no study in which cell polarity is successfully conferred to non-polar cells such as S2 cells in the cell autonomous manner. The three core components of the Par-complex are endogenously expressed in S2 cells but are distributed in the cytoplasm throughout the cell cycle. Thus, S2 cells appear to be an ideal system for cell polarity induction. We succeeded to reconstruct polarized Par-complex clustering by overexpressing Par3 in S2 cells. Using this polarity reconstruction system, we investigated the temporal pattern and dynamics of Par-complex clustering, and the fine structure of the Par-complex clusters at the super resolution level.

## Results

### S2 cells polarize by an elevated expression of Par3

First, we tested the effect of overexpressing each core component of the Par-complex in S2 cells, which distribute these components evenly throughout the cytoplasm and divide symmetrically (*Figure 1A*). We found that all core components of the Par-complex cortically co-localized in an asymmetric manner when Par3 was overexpressed, but did not cortically localize, when Par6 or aPKC was overexpressed (*Figure 1B,C*, and data not shown). We overexpressed myc-Par3 (or Par3-mKate2) via the *actin*-promoter (*act5c*)-driven *Gal4-UAS* system (*Act-Gal4 >UAS*) by transfection (see Materials and methods), with or without *actin*-promoter-*Par6-GFP* (*pAct-Par6-GFP*) as a live marker, which was uniformly distributed in the cytoplasm in the absence of Par3 overexpression (*Figure 1B*). Among transfected cells that exhibited cortical Par-complex distribution, a fraction exhibited an asymmetrically localized Par-complex (*Figure 1C*), while the rest of the cells localized uniformly to the cortex (see below). Asymmetric distribution of the Par-complex induced by Par3 overexpression required endogenous aPKC and Par6 (*Figure 1D*). The expression level of exogenous Par3 was also important for S2 cell polarization. Cortical polarization was not observed (*Figure 1B*) when Par3 expression, directly driven by the *actin* promoter, was approximately 1/40 of that of the *Act-Gal4 >UAS* system (*Figure 1E*).

We next examined the effect of endogenous Lgl that was largely localized uniformly along the cortex with a cytoplasmic distribution in S2 cells, prior to Par3 overexpression (*Figure 2A*). When Par3 was distributed asymmetrically along the cortex, Lgl and Par3 distributed in a complementary manner (*Figure 2B*). Knockdown of *lgl* via RNAi and the expression of Lgl3A, which aPKC is not able to phosphorylate, showed that Lgl and its phosphorylation by aPKC are required for asymmetric Par-

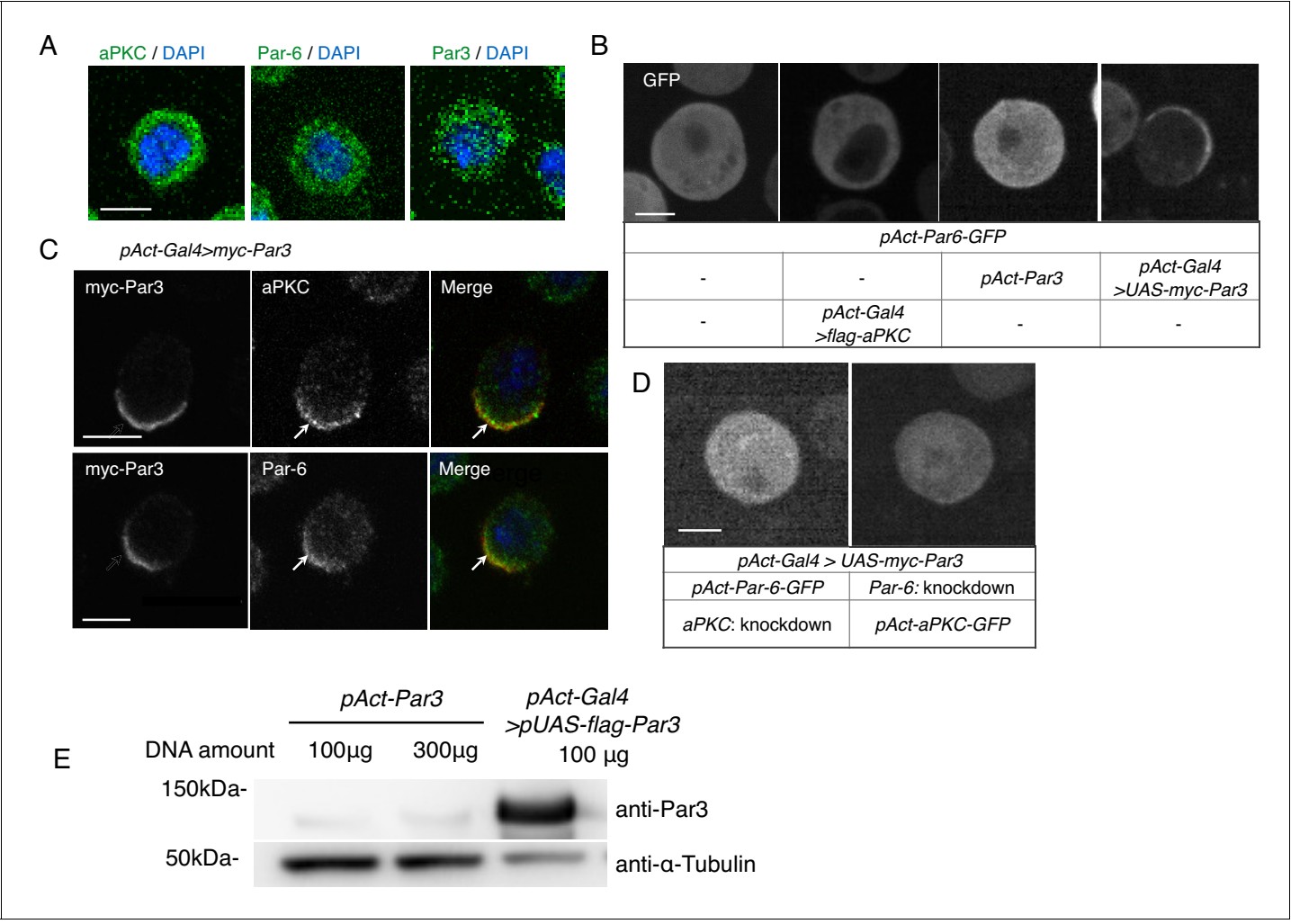

**Figure 1.** S2 cells polarize due to elevated Par3 expression. (A) Immunostaining of endogenous aPKC, Par6, and Par3 in S2 cells 2 days following transfection of the empty vector. Blue indicates DAPI staining. Images in A-D were at the equatorial plane of cells. Scale bar, 5 μm in all panels in this figure. (B) Live-imaging of Par6-GFP in S2 cells (top), 2 days following transfection of a combination of expression plasmids as described in the table (bottom). (C) Localization of endogenous aPKC and Par6 in cells overexpressing myc-Par3, stained with anti-myc-tag and anti-aPKC or anti-Par6 antibodies, and with DAPI, 2 days after transfection. Arrows indicate co-localized Par components. (D) Live-imaging of Par6-GFP (left) or aPKC-GFP (right) in Par3-overexpressing cells containing aPKC or Par6 RNAi knockdown, respectively, at 2 days post-transfection. (E) Comparison of the expression level of Par3-GFP driven by the *actin* promoter with that driven by the *actin*-promoter-*Gal4* x *UAS* system. Western blotting was performed for S2 cells transfected with *pAct-Par3-GFP* (100 μg and 300 μg/10⁶ cells) and with *pAct-Gal4* and *pUAS-Par3-GFP*, and the blot was stained with the anti-Par3 antibody to quantify the ratio of Par3-GFP amount driven by two different methods.
DOI: https://doi.org/10.7554/eLife.45559.002

complex localization in S2 cells (*Figure 2C,D*). We also confirmed that the other two components of Par-complex, Par6 and aPKC require *lgl* function to colocalize with Par3 along the cortex (*Figure 2E, F*).

To evaluate the degree of polarization of transfected cells, we introduced the asymmetric index (ASI), a measure of the polarized Par-complex distribution, which, according to *Derivery et al. (2015)*, indicates the degree of polarization of a fluorescent marker distributed along the equatorial cortex (*Figure 3A*). ASI distribution was compared with that of membrane-bound GFP (memGFP), which is essentially non-polarized (the control). The ASI value of memGFP ranged from 0 to 0.35 due to fluctuation. Par3 cortical distribution was categorized into two groups in comparison with memGFP. Cells with an ASI in the same range as that of memGFP were regarded as non-polarized. Those cells showing an ASI larger than that of the mem-GFP were interpreted as polarized. Among

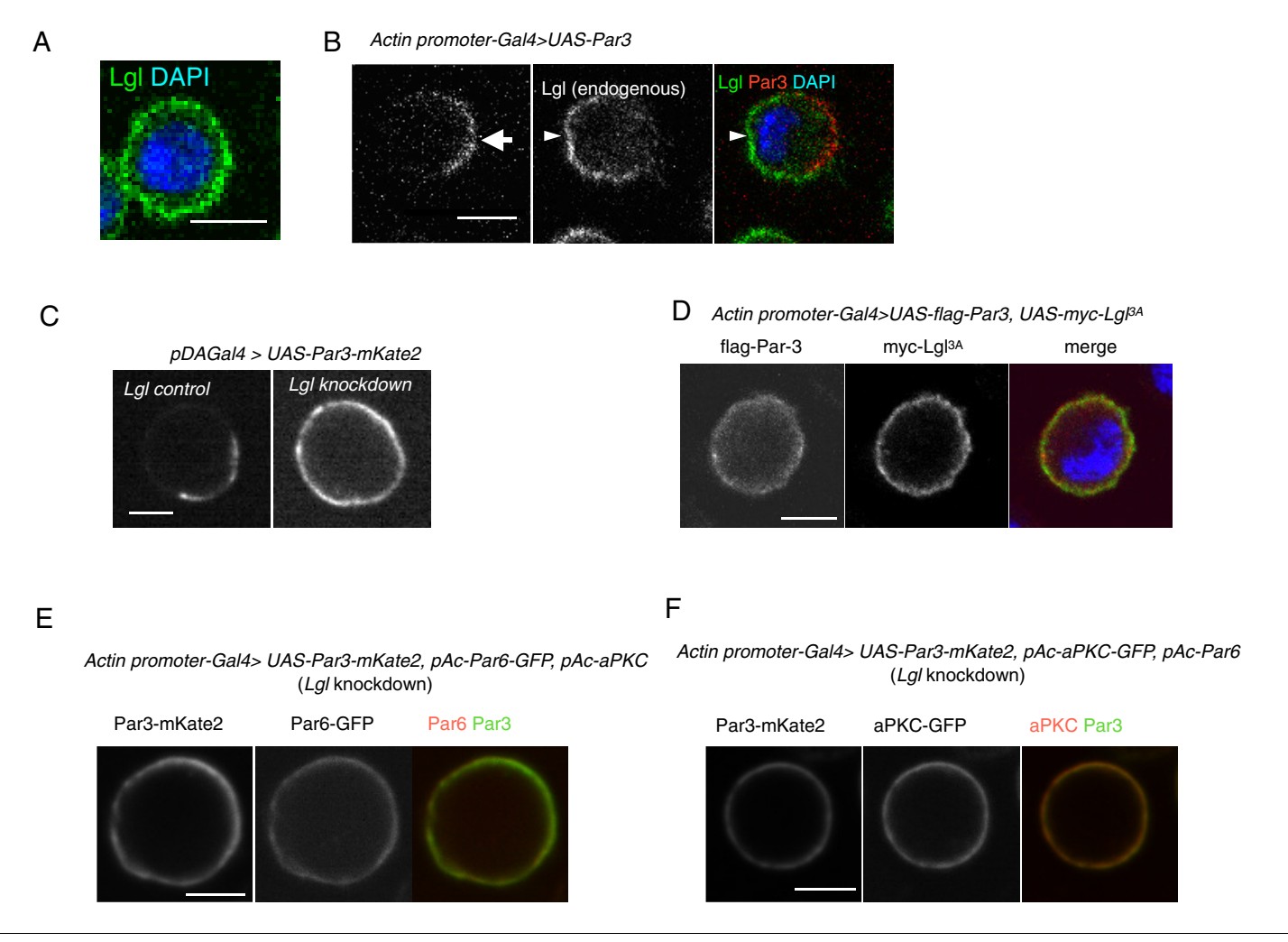

**Figure 2.** Par3 localization requires Lgl in S2 cells. (**A**) Endogenous expression of Lgl in S2 cells stained with anti-Lgl and DAPI at 2 days post-transfection of the empty vector. (**B**) Par3 and endogenous Lgl localize complementarily in 71% of cells (n = 24) where overexpressed Par3 was asymmetrically localized. Arrow, Par3 crescent. Arrowhead, Lgl. (**C**) Live-imaging of myc-Par3-mKates without (left) or with (right) Lgl knockdown by RNAi at 2 days post-transfection. (**D**) S2 cells over-expressing flag-Par3 and myc-Lgl3A, stained with anti-flag-tag, anti-myc-tag and DAPI. Lgl3A was cortically uniform in contrast to cytoplasmic Par3 distribution. (**E**) Live-imaging of myc-Par3-mKates and Par6-GFP with Lgl knockdown by RNAi at 2 days post-transfection. Myc-Par3-mKate2 co-localized with Par6-GFP in 100% of cells (n = 109) where the overexpressed Par3 was uniformly distributed. (**F**) Live-imaging of myc-Par3-mKates and aPKC-GFP with Lgl knockdown by RNAi at 2 days post-transfection. Myc-Par3-mKate2 co-localized with aPKC-GFP in 97.8% of cells (n = 137) where overexpressed Par3 was uniformly distributed.

DOI: https://doi.org/10.7554/eLife.45559.003

transfected cells showing cortical distribution of the Par complex, 39% were polarized, while 61% were non-polarized (*Figure 3B*).

Furthermore, we examined the three-dimensional (3D) distribution of the Par complex in S2 cells by reconstructing serial images at steady state 2 days following transfection. Interestingly, the region where the Par-complex accumulated was not uniform but consisted of multiple large aggregates (*Figure 3C*). These large aggregates were termed 'Par-islands'. These islands dynamically changed their arrangement on the surface of the S2 cells (*Video 1*).

We found that island structures of the Par-complex are also observed in the apical domain of *Drosophila* mitotic neuroblasts, where the Par-complex is asymmetrically localized (*Figure 3D*), suggesting that formation of Par-islands may be a common process during the polarized cortical distribution of the Par-complex irrespective of cell cycle phases.

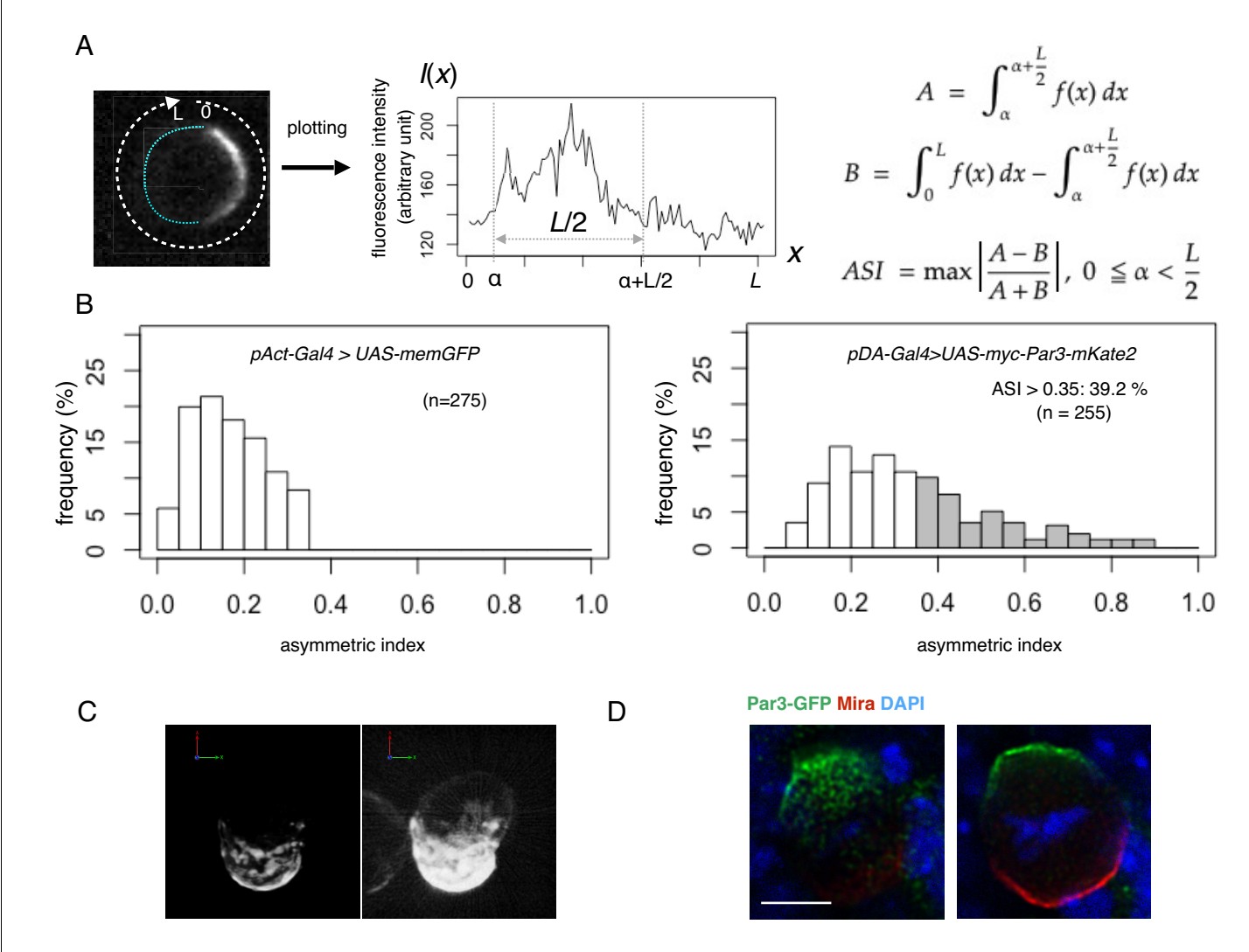

**Figure 3.** Quantification of cell polarity and 3D imaging. (**A**) Definition of the asymmetric index (ASI). ASI is defined by the maximum difference in the cumulative intensity of fluorescence (such as Par3-mKate2 and Par6-GFP) between a half cell perimeter and the other half at the equatorial plane of the cell, normalized via dividing by the cumulative intensity of the entire cell perimeter. (**B**) ASI distribution was compared between cells expressing memGFP and those expressing Par3-mKate2 at the cell cortex, both of which were driven by the Act-Gal4xUAS system. ASI value of memGFP is distributed broadly and ranges from 0 to 0.35. The mean value = 0.17 ± 0.08 (s.d.). In the all following figures, the numerical value after ± is s.d.. Since memGFP, in principle, has no ability to polarize, such a wide distribution originated from the fluctuation of random distribution along the equatorial cell perimeter and also from the existence of local membrane flairs. Distribution of ASI for cells showing cortical Par3 distribution (mean = 0.34 ± 0.18) may be categorized into two groups (**Figure 3A, B**); a group of cells show low ASIs overlapping with those of cells expressing memGFP in the range from 0 to 0.35, indicating that cells belonging to this group are essentially non-polarized. The ASI values of the other group (approximately 39%), are broadly distribute, but display ASI values larger than the ASI distribution of mem-GFP cells (ASI > 0.35, mean value = 0.52 ± 0.14). (**C**) 3-D reconstructed image of a cell overexpressing myc-Par3-mKate2 (left). In the right side image, brightness and contrast were adjusted to visualize the outline of the same cell. The time-lapse movie of a different cell is shown as **Video 1**. (**D**) Localization of the Par3-GFP in a mitotic neuroblast of a *Drosophila* brain expressing *Par3-GFP*, taken from a third instar larvae and stained for GFP (green) and Miranda (red, **Ikeshima-Kataoka et al., 1997**). The image were deconvolved. This neuroblast is tilted so that the apical pole (where Par3-GFP distributed) is on the near side, and the basal pole (where Miranda distributed) is on the far side. The left panel shows the image of a single focal plane near an apico-lateral surface. The right panel shows the image of the equatorial plane. Scale bar, 5 μm.

DOI: https://doi.org/10.7554/eLife.45559.004

The following source data is available for figure 3:

**Source data 1.** Source data for the histogram in *Figure 3*.
DOI: https://doi.org/10.7554/eLife.45559.005
**Source data 2.** The individual histogram in each experiment in *Figure 3*.
DOI: https://doi.org/10.7554/eLife.45559.006

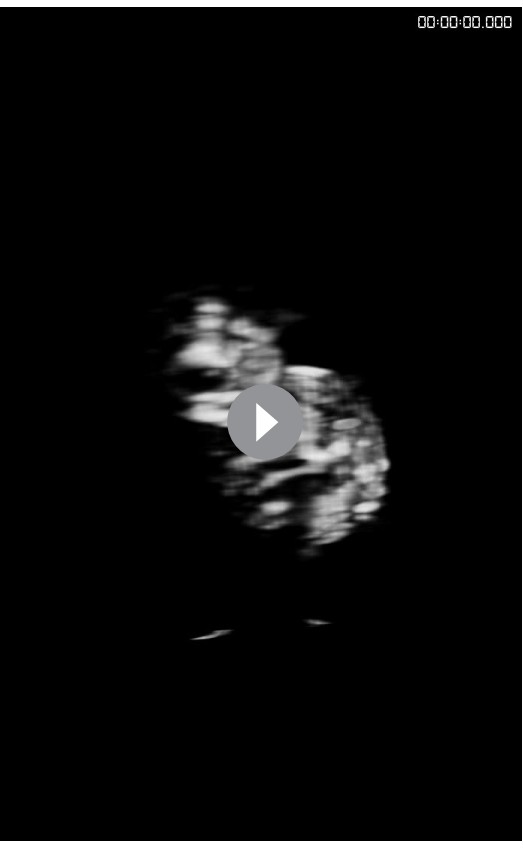

**Video 1.** 3D time-lapse movie of a polarized S2 cell monitored by Par6-GFP 2 days after transfection of pAct-Gal4 >UAS-myc-Par3-mKate2, pAct-Par6-GFP, and pAct-aPKC. Par-islands are clustered with dynamic movements.

DOI: https://doi.org/10.7554/eLife.45559.007

## Temporal patterns of Par-complex polarization

To investigate temporal patterns of polarized distribution of the Par-complex, we induced Par3 expression via the *Metallothionein* promoter, which is activated in the presence of $CuSO_4$. Because the expression level of Par3 shortly after induction might be too low to visualize, we designed the experiments to monitor the cortical Par complex formation with Par6-GFP that had been expressed by the *actin*-promoter. To test rationality of this experimental design, we transfected S2 cells with *pMT-myc-Par3-mKate2*, *pAct-Par6-GFP* and *pAct-aPKC*. Because expression of Par6-GFP and aPKC reached steady levels 2 days following transfection (*Figure 4—figure supplement 1*), induction of Par3-mKate2 was initiated at this time. When the distribution of Par6-GFP and Par3-mKate2 along the S2 cell cortex were examined at 3 hr post-induction, where small Par-islands have been formed (*Figure 4*), we confirmed that both proteins show a nearly identical cortical pattern in islands and dots (*Figure 4B,C,E*) with a minor difference in the local fluorescent intensity gradient between them. We also checked the relationship of this overlapping cortical distribution of Par3 and Par6 with that of Cdc42, which is known to form a complex with aPKC-Par6 separately from Par3-Par6-aPKC complex. We confirmed cortical distribution of Cdc42 in a punctate or short string shape (*Figure 4D*) (*Slaughter et al., 2013*; *Sartorel et al., 2018*). Cdc42 distribution appeared to be independent of those of Par3-mKate2 and Par6-GFP, and only a small fraction of Cdc42 dots resided at the edge of Par-islands (*Figure 4F,G*). Thus, we concluded that monitoring Par complex formation with Par6-GFP works properly. Using this monitoring system, we observed that the Par-complex first emerged in dot form (designated Par-dot) in the cell cortex during the initial phase of myc-Par3-mKate2 elevation, 2–3.5 hr following induction (*Figure 5A–C*, and *Video 2*), as well known in *C. elegans* one-cell embryos. Interestingly, Par-dots emerged in a restricted region of the cell cortex (*Figure 5C*).

Par-dots continued to grow in size via self-expansion, repeated fusion and less frequent fission (*Figure 5D*). These then developed into 'islands' of various sizes and shapes 2.5–6 hr following induction (*Figure 5A,B*), as observed in the *UAS-Gal4* system (*Figure 3C*). Par-islands dynamically developed by fusion with neighboring islands or dots as Par-dots did, and kept changing their shape and mutual position (*Figure 5E*). During this period, the distribution of Par-islands occurred via two separate processes, polar and non-polar clustering (*Figure 5A,B*), which corresponded with temporal changes in ASI values (*Figure 6A*). However, there was no significant difference between polarized cells and non-polar cells in either the time course of Par3-mKate2 expression levels or the Par6-GFP/Par3-mKate2 ratio (*Figure 6B–D*). While the average steady state amount of Par3-mKate2, driven by the *Metallothionein* promotor was approximately 1/16-fold of that driven by the *Act-Gal4 >UAS* system (*Figure 6E*, *Figure 6—figure supplement 1*), the appearance of the islands was similar between the two expression systems (*Figures 3C* and *5A*). The typical process toward Par-complex clustering is summarized in *Figure 6F*.

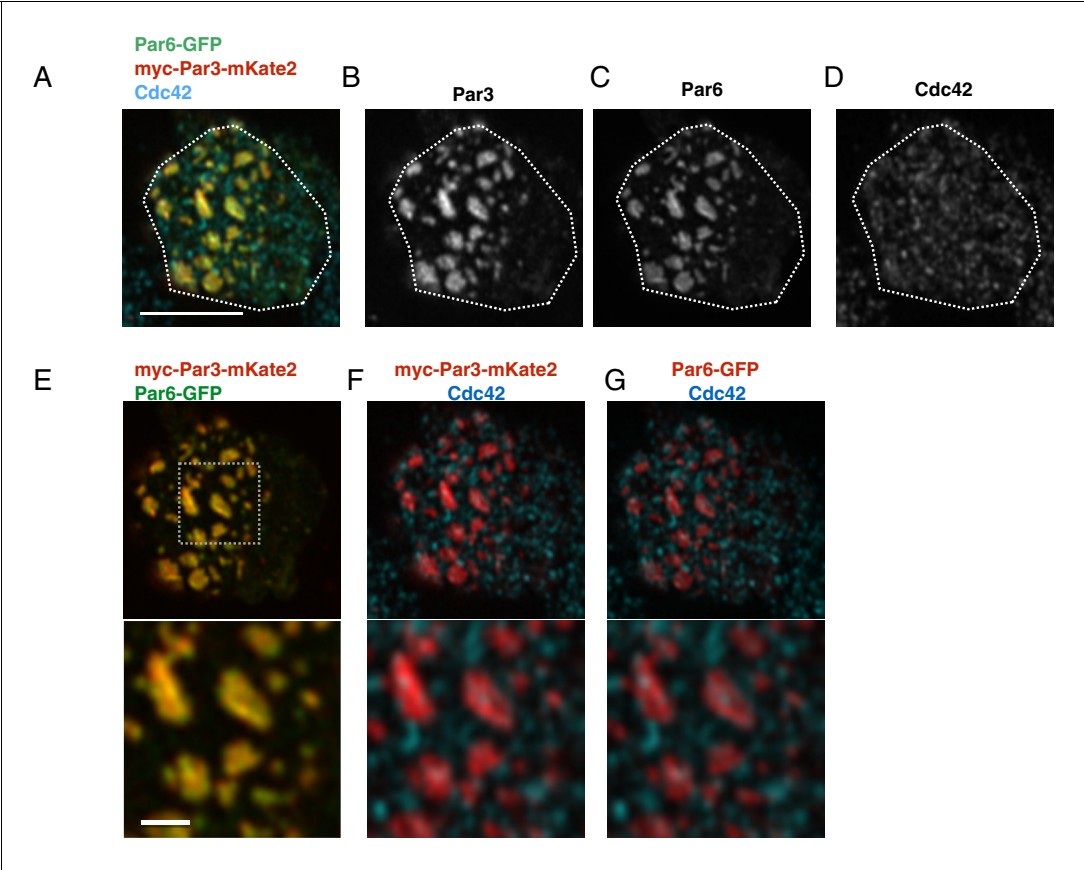

**Figure 4.** The localization of Par6, Par3 and Cdc42 in the Par-islands. (A–D) An image of a S2 cell expressing both myc-Par3-mKate2 and Par6-GFP, immunostained for myc, GFP and Cdc42. (A) Triple staining image for myc-Par3-mKate2, Par6-GFP and Cdc42. (B-D) Images showing each single immunostaining. The image was taken by focusing on a surface plane of the cell. Scale bar, 5 µm. (E–G) The double staining image of the cell shown in A-D. (E) myc-Par3-mKates (red) and Par6-GFP (green). (F) The image showing double staining for myc-Par3-mKate2 (red) and Cdc42 (blue). (G). The image showing double staining for Par6-GFP (red) and Cdc42 (blue). Lower panels of E-G show the magnified images of the dotted square. Scale bar, 1 µm. S2 cells are transfected with expression plasmids of *actin-promoter-Par6-GFP*, *actin-promoter-aPKC*, and *pMT-myc-Par3-mKate2*. Two days after transfection, $CuSO_4$ was administrated to induce myc-Par3-mKate2. Three hours post-induction, cells were fixed and immunostained.
DOI: https://doi.org/10.7554/eLife.45559.008
The following figure supplement is available for figure 4:

**Figure supplement 1.** Temporal pattern of Par6 expression level.
DOI: https://doi.org/10.7554/eLife.45559.009

## Dynamics at the steady state

Once the Par3-mKate2 expression level steadied about 8 hr following induction, approximately 30% of cells with Par6-GFP localized cortically demonstrated polarized distribution of Par6-GFP, resulting in a crescent in the equatorial plane (ASI > 0.35), while the rest of cells showed a non-polarized cortical distribution (ASI ≦ 0.35); (*Figure 7A*). At steady state, Par-islands dynamically change their mutual positions (*Figure 7B*, and *Video 3*). However, their asymmetric clustering and non-polar distribution were largely maintained for at least several hours, once cells reached steady state (*Figure 7C*), suggesting that both polar and non-polar clustering of Par-islands was fairly stable. In both states of Par-island clustering, the distribution of Par-islands and Lgl in S2 cells were mutually exclusive (*Figure 7D,E*, and *Videos 4* and *5*), suggesting that Lgl plays a role in the stability of the two states. Interestingly, Par-islands were never unified into one large island regardless of their dynamic movements as well as fusion and fission (*Figure 7B*).

Upon mitosis, cortical Par-islands disassembled and disappeared (*Figure 8A*). However, just prior to cell cleavage, small Par-dots reappeared preferentially in the region opposite the cleavage site

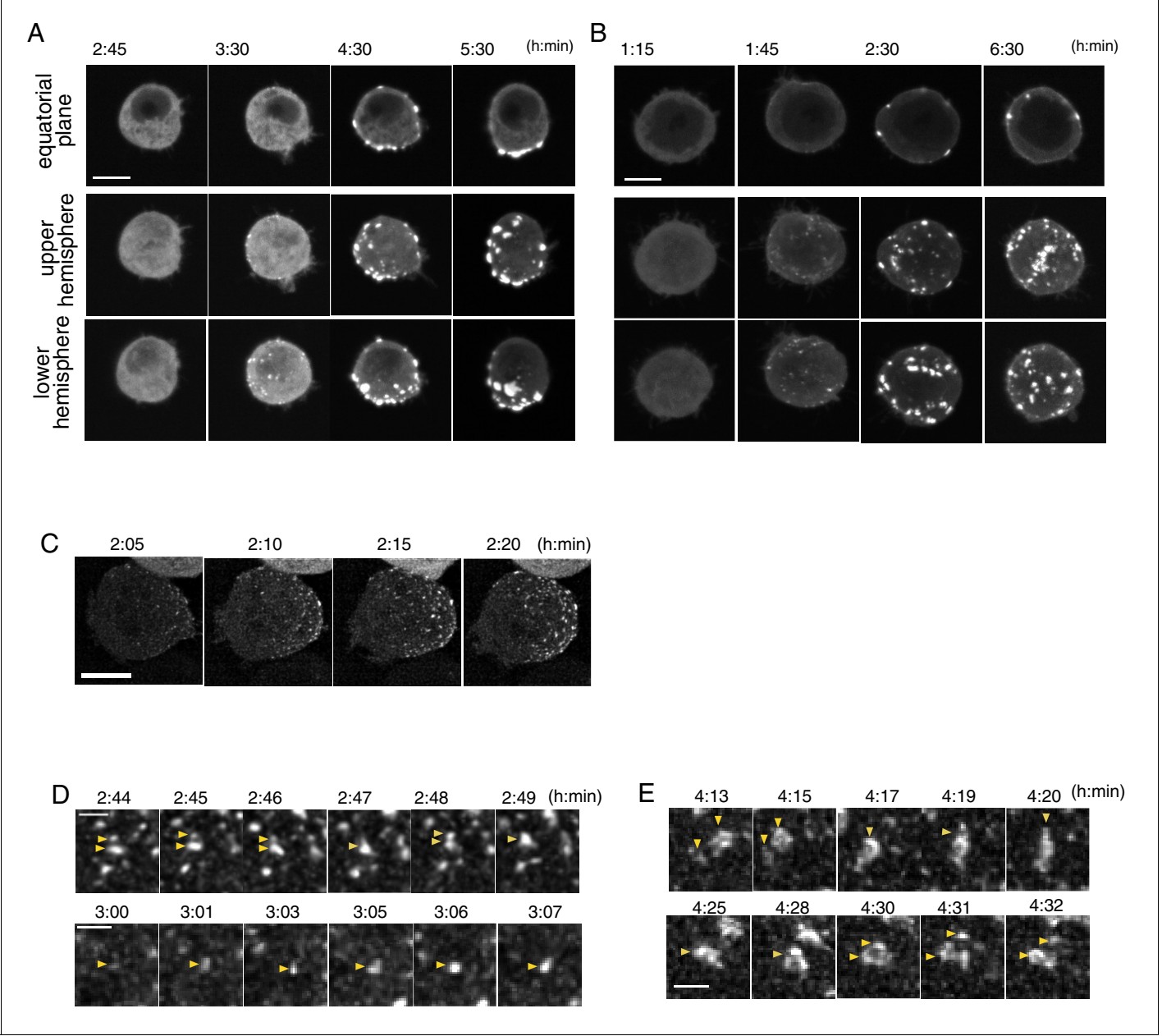

**Figure 5.** Temporal pattern of Par complex clustering. (**A and B**) Time-lapse images of S2 cells inducing Par3-mKate2 expression via the *Metallothionein* promoter, leading to polarized (**A**) or non-polarized (**B**) Par3 distribution. Time 0 (h: min) was at the time of induction by $CuSO_4$ addition in these and subsequent panels. The top row shows images at the equatorial plane. The middle and bottom rows show the max intensity projection images of the upper and lower hemispheres of the cell, respectively. Scale bar, 5 µm. (**C**) Time-lapse images of Par6-GFP showing the emergence and development of Par-dots. The images are 6 µm max intensity projection covering the entire cell. Scale bar, 5 µm. (**D**) Time-lapse imaging of Par6-GFP showing the fusion and fission of Par-dots (arrowheads in the upper panel), and the growth of a Par-dot (arrowheads in the lower panel). In E and F, scale bar, 1 µm. (**E**) Time-lapse image of Par-islands visualized by Par6-GFP. Arrowheads indicate dynamic shape changes, fusion (arrowheads, upper panel) and the dissociation of Par-islands (arrowheads, lower panel).

DOI: https://doi.org/10.7554/eLife.45559.010

(n = 11/14). Consistently, the position of the centrosome, which is normally located on the far side from the cleavage site, coincided weakly with the region of Par-dot reappearance (*Figure 8B–D*).

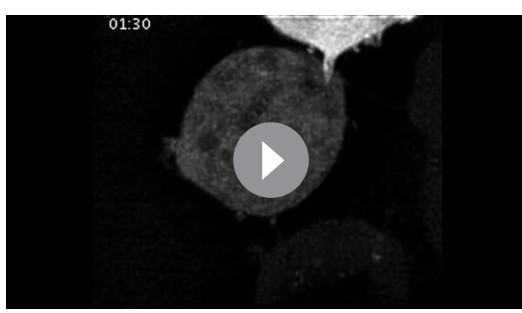

**Video 2.** 3D time-lapse movie of a S2 cell monitored by Par6-GFP following induction of myc-Par3-mKate2 from *Metallothionein* promoter. Induction started at time 0 by the addition of CuSO$_4$ 2 days after transfection of pMT-myc-Par3-mKate2, pAct-Par6-GFP, and pAct-aPKC.
DOI: https://doi.org/10.7554/eLife.45559.011

## Structural analysis of the assembly state of the Par-complex

A unique feature of the island form of the Par-complex was a slightly convex shape (*Figures 5A,B* and *7B*, and *Video 3*). To quantify the curvature of Par-islands in comparison with the membrane curvature of S2 cells, we utilized memGFP driven by the *actin*-promoter to visualize the cell membrane, together with myc-Par3-mKate2 induced by the *metallothionein*-promoter (*Figure 9A*). The images for quantification were taken at 8 hr after induction. The curvature radius for individual islands was obtained from three different points along the plasma membrane on individual islands in the equatorial plane of the cell. On the other hand, the curvature radius of the plasma membrane was calculated from the three different points along the cell membrane of 5 µm long. For this measurement, we chose the 'non-neighboring region' defined as the regions along the cell membrane more than 5 µm away from the edge of neighboring Par-islands. This is to avoid the influence of the presence of Par-islands (see Materials and methods section for details). As shown in *Figure 9B*, the curvature radii of Par-islands were in the range 0.7 µm to 5 µm with the median of 1.70 µm. (n = 69 islands in 34 cells). In contrast, the majority of curvature radii for the plasma membrane span in the range between 2 µm and 20 µm with the median of 4.07 µm in the non-neighboring regions (n = 87 plasma membrane regions in 29 cells). We also measured the curvature radii of non-Par-islands cells lacking Par-island formation, which gave us the median of 4.56 µm with the range of 2 µm and 20 µm (n = 76 regions in 13 cells). These results indicate that the curvature radius of Par-islands is significantly smaller than that of cell membrane curvature in non-neighboring regions, which is virtually consistent with the radius of S2 cells that we used for experiments. Accordingly, Par-islands show a more convex shape than the ordinary plasma membrane curve of S2 cells.

To better understand the organization of Par-islands, we investigated their structure at the super-resolution level. We performed super-resolution radial fluctuation (SRRF) analysis (*Gustafsson et al., 2016*), using confocal images of fixed samples, double-stained by Par3-mKate2 and Par6-GFP. This analysis revealed that both Par3-mKate2 and Par6-GFP exhibited polygonal shaped islands of various types (*Figure 10A,B*). In order to determine whether there was regularity in these structures, we measured contour lengths of the Par3-mKate2-stained meshwork, including separate rods and polygons. This measurement yielded distribution of lengths with multiple peaks in the density plot. We then searched the regularity of these multiple peaks. We found that it was well fitted with a combination of seven Gaussian curves, which exhibited a peak interval of 0.38 ± 0.06 µm (mean ± s.d., hereafter); (*Figure 10C,D,F*). We also performed spectral analysis of the density plot, and obtained a single major frequency of 2.4 µm$^{-1}$, which gives a peak interval of 0.42 µm (*Figure 10E*). These two analyses thus give consistent results with each other.

We also observed Par-islands of Par3-GFP and Par6-GFP separately via STED microscopy (*Figure 11A*) and found similar meshwork structures in the deconvoluted STED images (*Figure 11B–E*). The linear part of segments in the meshwork structures were measured (*Figure 11C*), and exhibited a length of 0.39 ± 0.09 µm (*Figure 11F*). Thus, these two methods essentially provided the same value for the segmental length. In addition, these segments had a fairly homogeneous diameter in STED microscopic images, where the mean half width of Par-segments was 0.22 ± 0.03 µm, (*Figure 11G,H*). These results raise the possibility that the Par-island meshwork contains a unit segment. Indeed, separate rod- or string-shape structures as well as open square structures were often observed in the earlier phases of the Par-complex aggregation time course (*Figure 11I*, and *Video 2*), supporting the notion that Par-islands are assembled from these elemental structures, generating regularity in the meshwork organization.

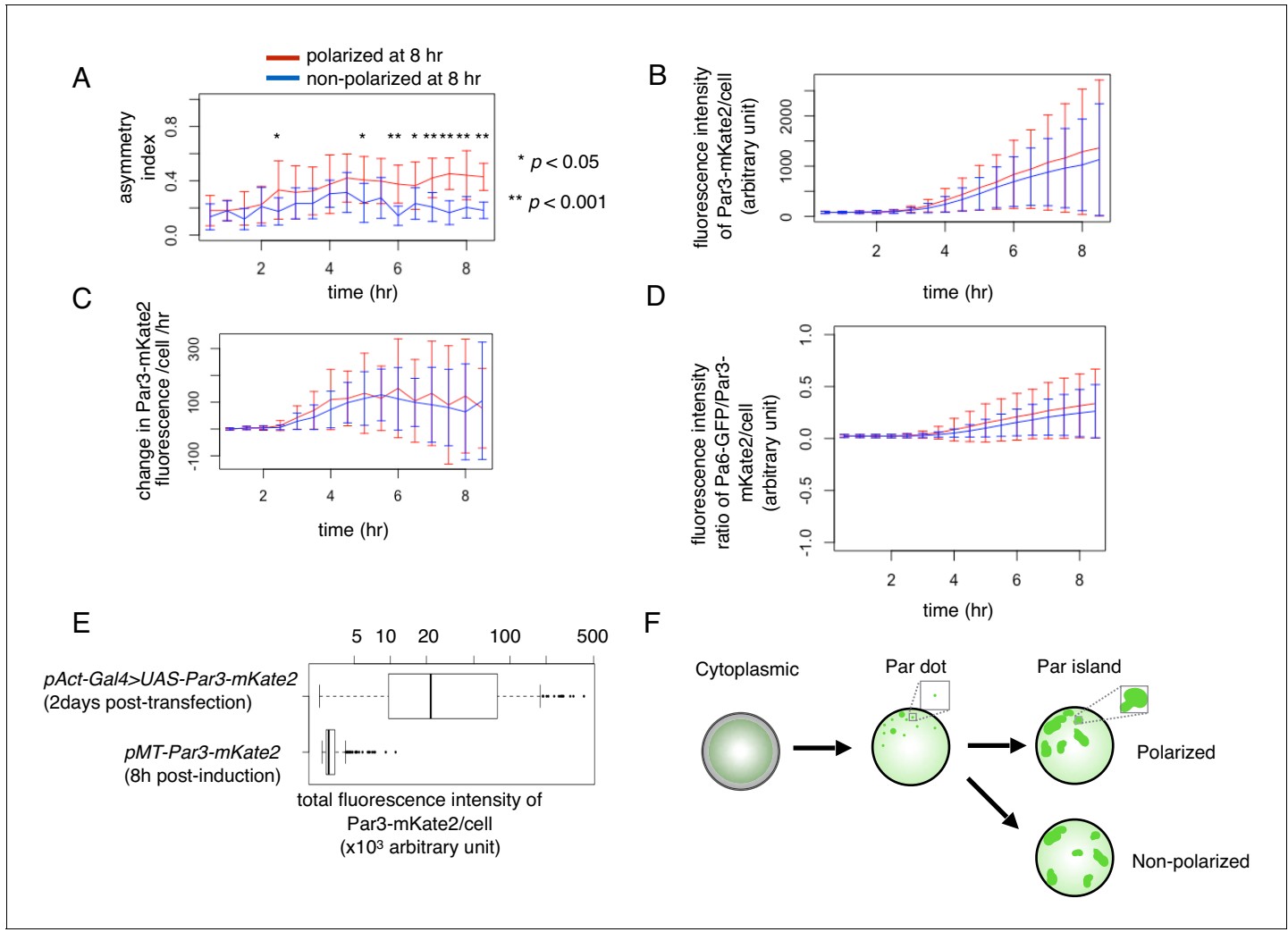

**Figure 6.** Temporal changes in cell polarity toward the two steady states. (A–D) Temporal pattern of the asymmetric index (A), fluorescence intensity of Par3-mKate2 (B), its rate of change (B) and the ratio of Par6-GFP/Par3-mKate2 (D) of S2 cells that were transfected with *pMT-Par3-mKate2* and *pAct-Par6-GFP*, followed by induction by $CuSO_4$ addition 2 days after plasmid transfection. Time 0 is the timing of $CuSO_4$ addition (2 days following plasmid transfection). Measurements were taken every 30 min. The blue and red line respectively indicates the averaged values of 10 cells showing a non-polarized Par6-GFP distribution (ASI around 0.2), and that of 13 cells with a polarized Par6-GFP distribution at 8 hr after induction (ASI around 0.4). Fluorescence intensity reached the steady level around 8 hr after induction (B, D). Bars indicate s.d. The temporal pattern of the fluorescence intensity/cell of Par3-mKate2 and Par6-GFP/Par3-mKate2 ratio are not significantly different between the polarized cell group (red line) and non-polarized cell group (blue line). The ASI value began to increase immediately after the rise of Par3-mKate2 levels (approximately 2 hr after induction) in the polarized cells (blue line) in (A), and maintained a high level afterwards, while the non-polarized cells (red line) initially showed a slight increase in the ASI value and subsequently a decrease from 5 hr after induction onwards. The timing of the increase in ASI roughly corresponded to that of Par-dot emergence (2–4 hr after induction; see *Figure 4A–C*), and the timing of a decrease in ASI value in non-polarized cells roughly corresponds to the late period of Par-island formation (4–6 hr after induction), although there are cell-to-cell variations in these timings. in (A), *t* test, p=$5.6×10^{-5}$, 0.04, $5.5 × 10^{-4}$, $1.1 × 10^{-6}$, $5.5 × 10^{-4}$, and $4.3 × 10^{-7}$ for every 30 min time point from 6 hr after induction. (E) Comparison of Par3-mKate2 expression level induced by the *Metallothionein* promoter, *pMT-Par3-mKate2,* and that promoted by the *pAct-Gal4xUAS* system. The mKate2 fluorescence intensity of the individual S2 cells was measured 2 days after transfection of *pAct-Gal4 and UAS-Par3-mKate2*, or at 8 hr post-$CuSO_4$ induction of *pMT-Par3-mKate2,* 2 days after transfection of the plasmid. The expression level of Par3-mKate2 per cell was approximately 16-fold higher when it was driven by the *UAS-Gal4* system (Mean $5.5 × 10^4 ± 7.1 × 10^4$) than that of the steady state level induced by the *Metallothionein* promoter ($3.4 × 10^3 ± 9.8 × 10^2$). We also estimated the ratio of overexpressed Par3 protein level to endogenous Par3 in S2 cells to be approximately 300-fold and 20-fold for the *Gal4-UAS* system and *Metallothionein* promoter, respectively (*Figure 6—figure supplement 1*). (F) Schematic presentation of S2 cell polarization process from Par-dot formation to clustering of Par-islands.

DOI: https://doi.org/10.7554/eLife.45559.012

The following source data and figure supplement are available for figure 6:

**Source data 1.** Source data for the temporal change of S2 cells.

*Figure 6 continued on next page*

*Figure 6 continued*

DOI: https://doi.org/10.7554/eLife.45559.014

**Figure supplement 1.** Comparison in the expression level between the endogenous Par3 and the induced Par3.

DOI: https://doi.org/10.7554/eLife.45559.013

## Roles of Par components and the cytoskeleton in polarity formation

Because the elevation of Par3 expression induced cortical polarization in S2 cells, we investigated the role of functional domains of Par3 by observing phenotypes with Par6-GFP following the overexpression of mutant *Par3* forms via the *Metallothionein* promoter (*Figure 12A*). First, we tested the role of the CR1 domain responsible for self-polymerization in the polarized Par-complex assembly (*Benton and St Johnston, 2003*; *Mizuno et al., 2003*).

Overexpression of Par3 lacking CR1(Par3ΔCR1) in the presence of the endogenous Par3 compromised the cortical Par-complex assembly significantly. The Par-complex was broadly distributed over the cell cortex in the initial stages, when dots were very faintly visible. Although brighter fluorescence spots (similar to Par-islands) formed later, they were relatively smaller than the Par-islands in size, and mostly faint with ambiguous contours, compared to those formed by wild-type Par3 expression (*Figure 12B*). Eventually, distribution of these Par-aggregates was little polarized (*Figure 12C*). These results suggested that the CR1 domain was important for all processes during the development of macro-scale structures of the Par-complex.

However, the endogenous Par3 expression may contribute to this phenotype under this condition. Therefore, we next examined the effect of Par3ΔCR1 overexpression after knocking down the endogenous Par3 by RNAi (*Figure 12—figure supplement 1*). RNAi treatment reduced the endogenous Par3 expression to 10% of the normal Par3 expression level. As the next step, we introduced silent mutations into the RNAi sequence part of the full-length Par3 and Par3ΔCR1 in the overexpression constructs to prevent the expression of these exogenous proteins from being knocked down by RNAi. We confirmed that the expression levels of the exogenous proteins bearing silent mutations, myc-Par3sm-GFP and flag-Par3ΔCR1sm-GFP, were not affected by simultaneous RNAi treatment of cells, and that the expression level of both exogenous proteins were nearly at the same level (*Figure 12—figure supplement 2*).

We first examined the distribution of myc-Par3sm-GFP and flag-Par3ΔCR1sm-GFP by STED super-resolution microscopy. We confirmed a various size of Par-islands including large ones with a regular meshwork structure when full-length myc-Par3sm-GFP was overexpressed (*Figure 12D*, compare *Figure 11B*). On the contrary, in the cells overexpressing flag-Par3ΔCR1sm-GFP, we often observed bright spots with an amorphous shape, and could rarely find clear island structures with a regular meshwork structure (*Figure 12E*). Thus, our STED observations strongly suggest that in the absence of Par3 CR1 domain, Par-islands consisting meshworks with unit-like segments were not formed.

We also measured the size of islands in S2 cells expressing Par3sm and that of bright regions of Par3ΔCR1sm at the steady state after RNAi suppression of the endogenous Par3 (see Materials and Materials and methods for details). The results show that the distribution of spot size after Par3ΔCR1 overexpression, which was independent of the knockdown of endogenous Par3 (data not shown), was quite different from that of Par-islands formed by the overexpression of Par3 (*Figure 12F,G*); approximately 90% of bright spots are smaller in area than 1 $\mu m^2$ (the mean value = $0.44 \pm 0.37$ $\mu m^2$ (s.d.)). The size distribution of bright spots of Par3ΔCR1 was indistinguishable from most bright spots of Par-dots 2.5 hr after myc-Par3-GFP induction, when only a small fraction becomes Par-islands (compare *Figure 12G and H*). These observations raise the possibility that Par3ΔCR1 still has an ability to form some cortical aggregates, either via (1) an un-identified domain in Par3 or (2) via Par6-aPKC, or (3) via the other components, while the development of Par3ΔCR1-containing aggregates is very limited.

All together, we conclude that the CR1 domain of Par3 is critical for Par-complex to form Par-islands with the regular meshwork and to asymmetrically localize in S2 cells. This conclusion is consistent with the previous finding that CR1 domain is an essential domain of the Par3 function to form the normal asymmetry in the cell cortex in the epithelium (*Benton and St Johnston, 2003*).

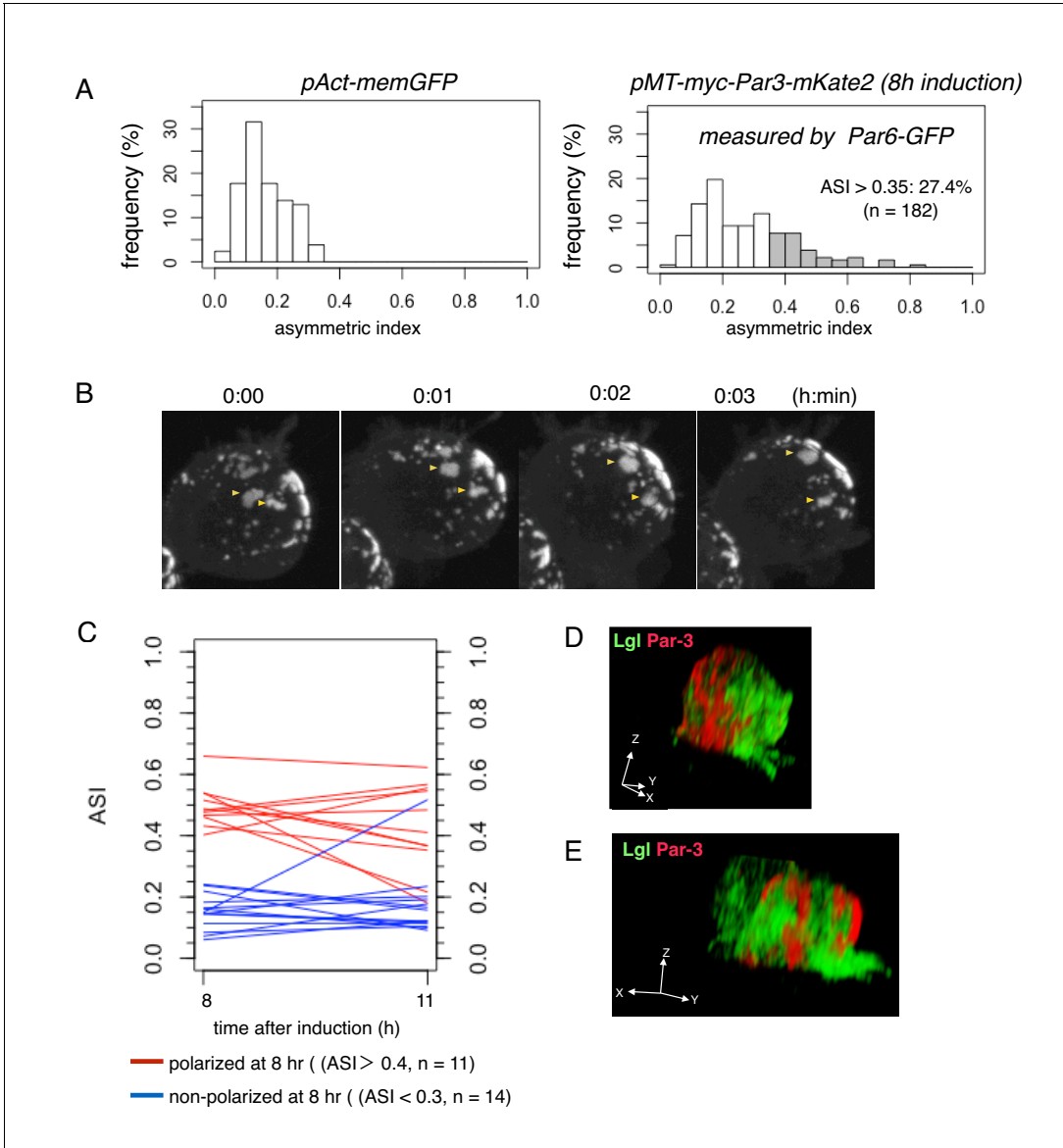

**Figure 7.** Steady state dynamics of polarized Par complex. (**A**) The distribution of ASI among cells with memGFP (left) driven by the *Act5C* promoter and Par-3-mKate2 (right) induced MT promoter. ASI was measured for the equatorial plane of cells 8 hr after CuSO$_4$ addition. The mean ASI value was 0.17 ± 0.08 (s.d.) for cells expressing memGFP (n = 209 cells), and 0.27 ± 0.15 (s.d.) for cells localizing Par3-mKate2 along the cell cortex (n = 182 cells). Cells showing ASI in the range outside the ASI distribution for memGFP expressing cells (ASI > 0.35) were 27.4% of the cells with cortical Par3-mKate2 (approximately 52% of the transfected cells localize mKate2 cortically). Mean ASI value for those cells was 0.43 ± 0.12 (s.d.). In all figures and the main text, s.d. is shown following the mean value. (**B**) Time-lapse imaging of Par-islands at the steady state, taken 8 hr after the induction of Par3-mKate2 by CuSO$_4$ addition and onwards. Par-islands in a polarized cell were visualized by Par6-GFP that had been expressed by the *actin*-promoter for 2 days prior to Par3-mKate2 induction. See also *Video 3*. (**C**) Stability of polarized and non-polarized cells. ASI values 11 hr post-induction were measured for cells polarized 8 hr post-induction (ASI > 0.4, n = 11) and for non-polarized cells (ASI < 0.3, n = 14). ASI values were measured using induced Par6-GFP. (**D and E**) 3D images of the distribution of myc-Par3-mKate2 and endogenous Lgl in cells showing polarized (**D**) and non-polarized (**E**) Par3 localization. The distribution of Lgl and myc-Par3-mKate2 is essentially non-overlapped in both cases. See *Videos 4* and *5* for the 3D-rotation movies.

DOI: https://doi.org/10.7554/eLife.45559.015

The following source data is available for figure 7:

**Source data 1.** Source data for the histogram in the steady state.
DOI: https://doi.org/10.7554/eLife.45559.016

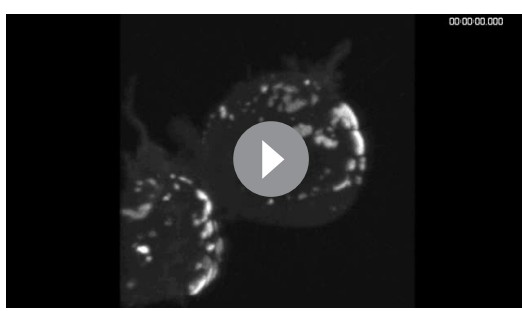

**Video 3.** 3D time-lapse movie of a polarized S2 cell monitored by Par6-GFP at 8 hr induction of myc-Par3-mKate2 with the co-expression of pAct-Par6-GFP and pAct-aPKC.

DOI: https://doi.org/10.7554/eLife.45559.017

We next examined the effect of aPKC-dependent phosphorylation at Serine 980 in the CR3 domain, which is necessary for dissociation of Par3 from aPKC (*Figure 13A*) (*Morais-de-Sá et al., 2010*). Overexpressing the non-phosphorylatable form, Par3S980A, which tightly binds aPKC (*Morais-de-Sá et al., 2010*), rapidly promoted aPKC complex aggregation, and the polarized region initially assumed a bowl-like shape, in which clustering of the Par-islands was so tight under this condition that the island structure was not easy to be discriminated (see images at 3 hr 20 min in *Figure 13B*). This dense aggregation gradually separated into small and nested islands, and eventually resulted in an increase in the polarized cell population (4 hr 20 min in *Figure 13B*, and *Figure 13C*); 40% of cells with cortical Par3 showed an ASI > 0.35, and a degree of polarization with a mean ASI value of 0.52 ± 0.13 for polarized cells (*Figure 13C*). The initial dense packing of the Par-complex containing Par3S980A suggested that the turnover of Par3-aPKC association and dissociation might play a role in the normal clustering of Par-islands. This was similar to that of *Drosophila* epithelial cells, wherein Par3S980A colocalized with aPKC-Par6 in the apical domain with disorganized adhesion belts (*Morais-de-Sá et al., 2010*).

Next, we examined the effect of the membrane association region (MAR) of Par3 by overexpressing Par3ΔMAR (*Krahn et al., 2010*). The Par-complex no longer localized cortically, but formed several cytoplasmic aggregates, which coalesced into a single large sphere (*Figure 13D*) as previously observed in the oocyte (*Benton and St Johnston, 2003*). Thus, the functional domains of Par3 and the interactions between these domains, together, play a role in the properly polarized distribution of the Par-complex in the S2 cell system.

Lastly, we examined the effects of the actin cytoskeleton on islands. While ROCK inhibitor, Y27632, did not significantly affect the behavior of Par-islands (data not shown), an actin inhibitor, Latrunculin B, changed the islands into a spherical shape, which frequently formed membrane protrusions (*Figure 13E*, and *Video 6*), suggesting that the actin-membrane skeleton is necessary to balance the surface tension of Par-islands (see Discussion).

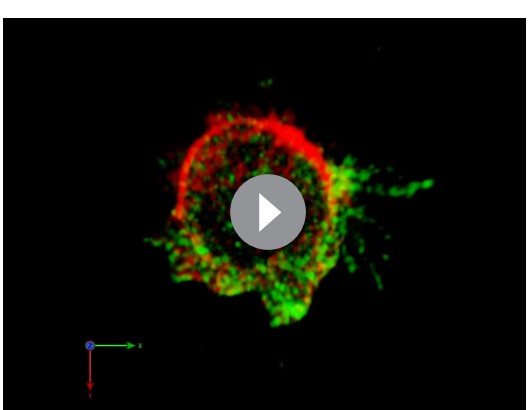

**Video 4.** 3D movie of a polarized S2 cell stained for myc-Par3-mKate2 and Lgl at 8 hr induction of myc-Par3-mKate2 with the co-expression of pAct-Par6-GFP and pAct-aPKC.

DOI: https://doi.org/10.7554/eLife.45559.018

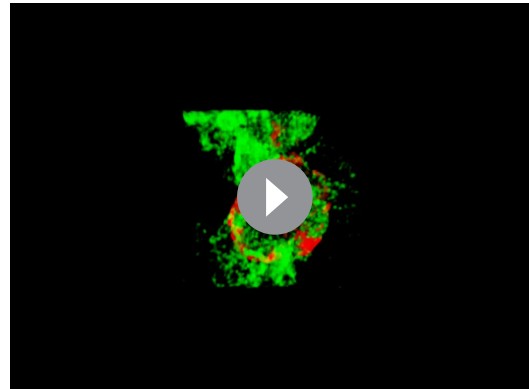

**Video 5.** 3D movie of a nonpolarized S2 cell stained for myc-Par3-mKate2 and Lgl at 8 hr induction of myc-Par3-mKate2 with the co-expression of pAct-Par6-GFP and pAct-aPKC. A part of the adjacent cell is included in the movie.

DOI: https://doi.org/10.7554/eLife.45559.019

## Discussion

In this study, we were successfully able to reconstruct Par-complex-dependent cortical cell polarity induced by Par3 overexpression in non-polar *Drosophila* S2 cells, using the *Gal4-UAS* system and the *Metallothionein* promoter for Par3 expression. Using this polarity reconstruction system, we revealed that large Par complex clusters are composed of a meshwork containing unit-like segments, which dynamically associated and dissociated with each other. S2 cells endogenously express Par3, Par6 and aPKC, but never cortically localize them nor asymmetrically in the natural state. The cell polarity reconstructed by our strategy requires endogenous Par6, aPKC, and Lgl, and hence this reconstruction system appears to reproduce the fundamental properties of Par-dependent polarization in vivo, at least in part. Especially, the apical crescent of Par-complex in *Drosophila* neuroblasts turned out to have a large structure similar to Par-island structures that evolves in S2 cell system. This would be another evidence that this reconstruction is useful model in order to extract general properties of polarization of the Par-complex. At the same time, we recognized there are some differences between the current reconstruction system and the endogenous polarity formation. We will discuss the advantages and limit (or points to be improved) of this system.

### Temporal patterns of Par-complex aggregation

In our reconstruction system, cortical asymmetry began with the formation and growth of cortical dot-like structures, which were also reportedly associated with anterior localization of the Par-complex in *C. elegans* zygotes (*Wang et al., 2017*; *Dickinson et al., 2017*; *Munro et al., 2004*). Par-dots in the S2 cell system included all three Par-complex components. Thus, these dots appear to be the common initial process of Par-complex cortical aggregation. In parallel with the growth of Par-dots, string-like structures often emerged. The subsequent process of asymmetric localization proceeded in the form of Par-islands with amorphous and dynamic behavior. To our knowledge, this type of Par-complex form with such a dynamic behavior has not been explicitly described in cortical Par-complex assembly in *C. elegans* one-cell embryo or *Drosophila* neuroblasts. We indeed found the island-like structures in the Par complex distribution in *Drosophila* mitotic neuroblasts (*Figure 3D*). We also realized that a similar behavior of Par aggregates is detectable in the movies of *C. elegans* one cell embryo, whereas it is not explicitly described (*Wang et al., 2017*). Furthermore, Par3 is known to show punctate localization at the early stage of *Drosophila* epithelialization (*Harris and Peifer, 2005*; *Harris and Peifer, 2007*). Par3 is also localized as a small patch form on the niche side of interphase female germline stem cells in *Drosophila*, while its behavior is unknown (*Inaba et al., 2015*). Based on these, we suggest that Par-island structure is not specific to this artificial apolar S2 cells but a universal form of Par-complex aggregates.

Interestingly, initial dot formation appeared to be biased toward the region opposite the cleavage point, where the centrosome also appeared to be located, which was consistent with a recent study on *Drosophila* (*Loyer and Januschke, 2018*; *Januschke and Gonzalez, 2010*; *Jiang et al., 2015*). Thus, the cleavage point and/or the centrosome may be a general positional cue for the initiation of Par-complex-dependent cell polarity. In this context, polarization process of the S2 cell system is likely to be cell-autonomous and dependent on the induction of polarity proteins, wherein the orientation of polarity appeared to be dependent on internal cue(s).

### The difference between S2 cell reconstruction system and *Drosophila* neuroblasts

Whereas our synthetic polarity shares a similar meshwork organization of Par-complex aggregates with *Drosophila* mitotic neuroblasts, there are two major differences between them: (1) cell cycle phase at which polarity is formed and (2) time constant of polarity formation (see the following section 'Dynamic behaviors of Par aggregation and mechanisms for clustering' for the time constant). In *Drosophila* neuroblasts, Par-polarity starts to form G2 phase and mainly operates in the mitotic phase, disappearing once they enter G1 phase. In contrast, in our artificial polarity system of S2 cells, Par-polarity is formed during interphase, and disappears upon mitotic entry. This difference in cell cycle phase of polarization affects the spatio-temporal relationship of Par-complex and other components; Lgl distributes cortically throughout interphase and prophase but relocalizes to the cytoplasm during prophase. Thus, Lgl cortically distributes when cell polarity is artificially formed in our S2 system, hence their complementary distribution lasts as long as Par-complex cortically distributes

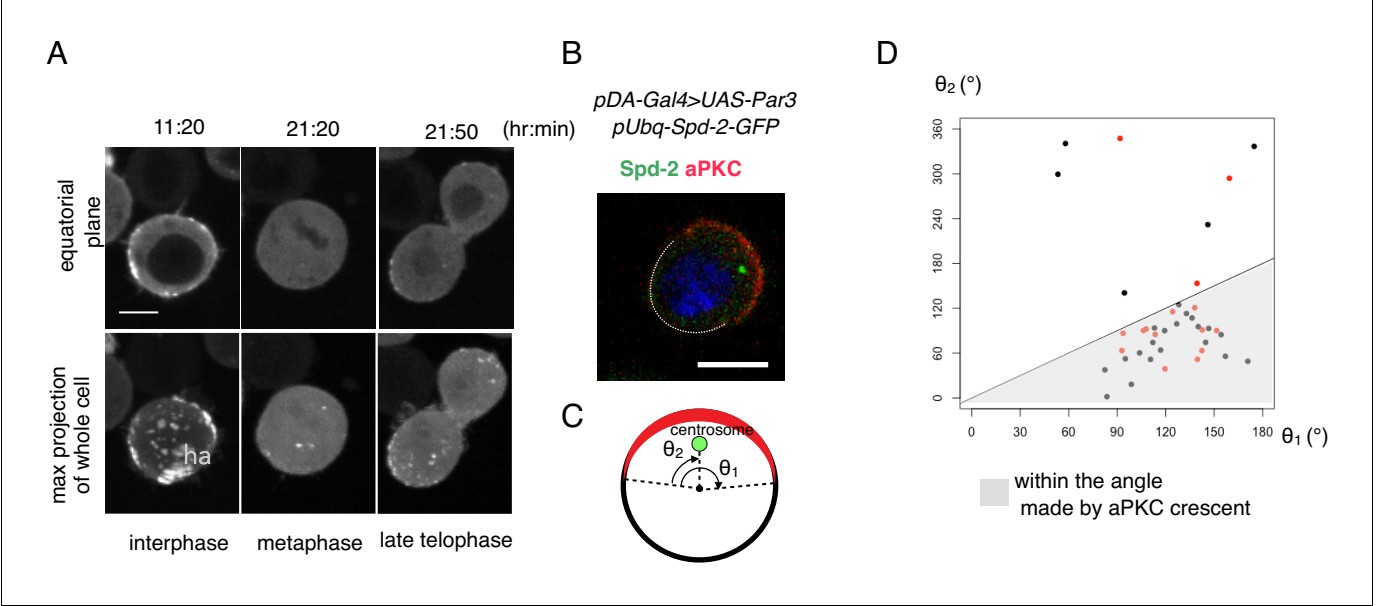

**Figure 8.** Cell autonomous formation of the Par complex polarization. (**A**) Time-lapse imaging of Par3 distribution during mitosis. Time indicates hr:min after CuSO$_4$ addition. Images of equatorial plane (upper panels), and the max projection of the whole cell (lower panels) are shown. Scale bar, 5 μm. (**B–D**) Relationship between the Par-complex crescent and the position of the centrosome. The centrosome was visualized via Spd-GFP, which was expressed by the transfection of *pUbq-Spd-2-GFP* and *pDA-Gal4* together with *pUAS-Par3*. Spd-2-GFP and aPKC were immunostained (**B**). The radial angle the aPKC crescent from the cell center (θ1) and the angle between an edge of the crescent and the centrosome (θ2) in the clock-wise direction were measured at the equatorial plane (**C**). In 32 out of 40 cells (80%, the sum of independent experiments), the centrosome was located within the fan shape made by the aPKC crescent and the cell center (**D**). Data shown in black dots and red dots were obtained by immunostaing and live-imaging, respectively.

DOI: https://doi.org/10.7554/eLife.45559.020

The following source data is available for figure 8:

**Source data 1.** Source data for the position of centrosome.
DOI: https://doi.org/10.7554/eLife.45559.021

(*Figures 2* and *7D,E*). In contrast, in neuroblasts, their complementary distribution occurs only in a short period within the mitotic phase when Par complex localizes asymmetrically in neuroblasts, because Lgl rapidly relocalizes to the cytoplasm during prophase.

As for the disappearance of the cortical Par complex at mitosis, our artificial polarity system might resemble epithelial cells, in which apical components including aPKC, Par3 and Crumbs disappear from the apical side during the mitotic phase (*Bergstralh et al., 2013*). Neuroblasts that lose adherens junctions during delamination from epithelial layer must acquire a specific mechanism to maintain Par-polarity during mitosis. The Par-complex is activated by AuroraA kinase (*Wirtz-Peitz et al., 2008*). A possibility is that different levels of such mitotic kinases including AuroraA and Polo kinase may cause the disappearance of Par-complex clustering at mitosis in the S2 cell system.

## The morphology and dynamics of Par-islands

Par-complex assembly at the cortex of S2 cells appears to stabilize the cell membrane because membrane filopodia extensively formed in areas where Par-islands were absent (*Video 2*). Also, cell membrane curvature was higher where Par-islands were attached, compared with that of the surrounding areas (*Figures 5A,B* and *9B*). Membrane curvature may be determined by the balance between elasticity of the cortical cytoskeleton, the affinity of the Par-complex toward the cell membrane, and possibly the surface tension of the Par-island. Membrane affinity of the Par-complex is mediated by Par3 MAR, which interacts with phosphoinositides (PIPs) (*Krahn et al., 2010*) and/or by Par6-cdc42 interaction (*Joberty et al., 2000*). The convex shape of the Par-island and its higher membrane curvature reflects its relatively high surface tension. This is supported by the fact that disruption of the actin cytoskeleton by Latrunculin B treatment leads to a curled or spherical Par-island,

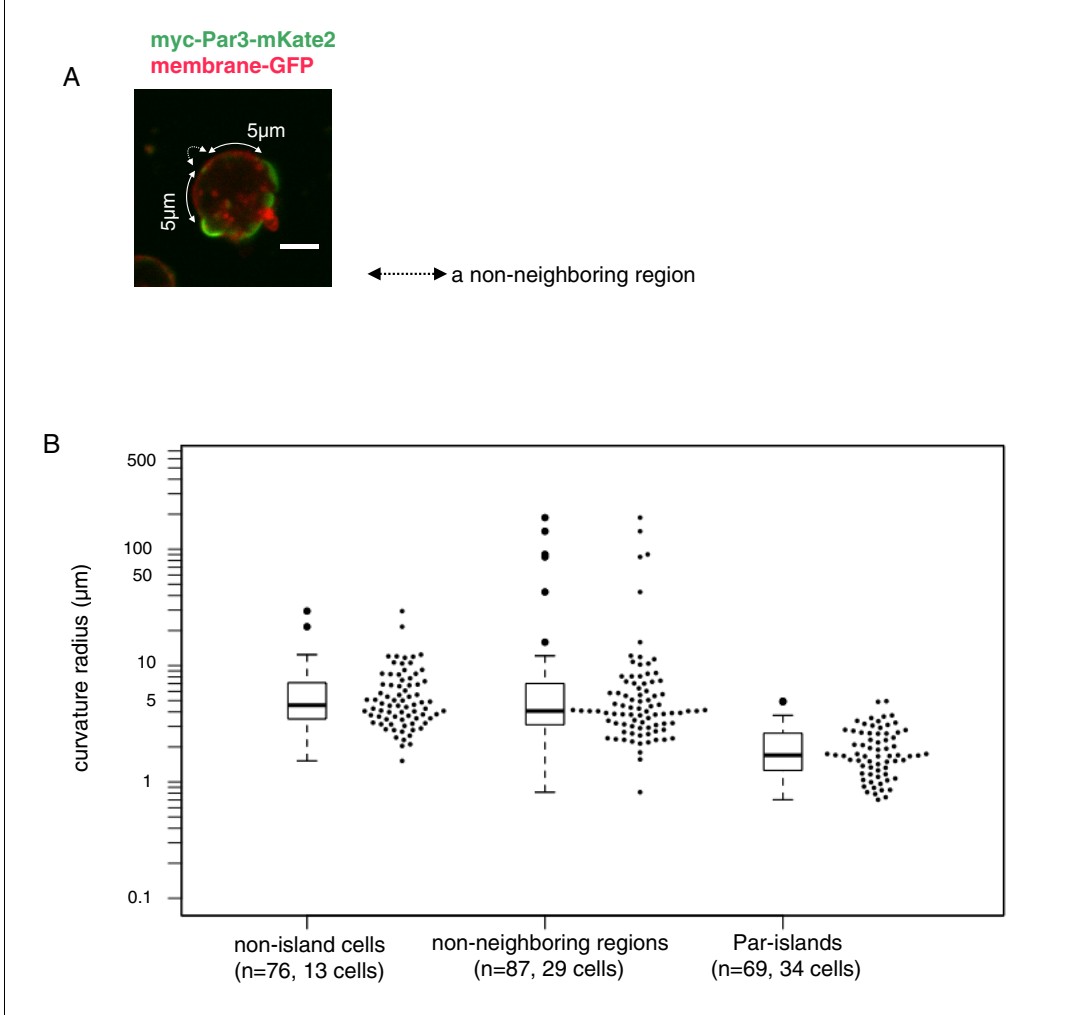

**Figure 9.** Curvature of Par-islands. (**A**) Cell membrane regions along the equatorial contour were visualized by memGFP and classified into three regions, Par-islands, neighboring regions, and non-neighboring regions. A Par-island region was defined as the bright mKate2 fluorescence region where a Par island crossed the equatorial plane, with over 1 µm in length along the equatorial contour. A 'non-neighboring region' was defined as a region more than 5 µm away from an edge of Par-islands along the cell contour. In each Par-island and non-neighboring membrane region, the three coordinates, both edges and a mid point along the memGFP contour were selected, and the curvature radius was calculated by the values of coordinates of the three points. Non-islands cells are those lacking Par-islands (see Materials and methods for details). The live-imaging data were taken 8 hr after $CuSO_4$ addition. In most cells, both Par-islands and non-neighboring membrane regions were able to be selected from each single cell. Scale bar, 5 µm. (**B**) Curvature radii of Par islands, non-neigboring membrane region of cells bearing Par-islands, and of non-island cells. The median of curvature radii are 4.56 µm for non-island cells (n = 76 for 13 cells), 4.07 µm for the non-neighboring-region (n = 87 for 29 cells) and 1.70 µm for the Par-islands (n = 69 for 34 cells). The median curvature radius of Par-islands is significantly smaller than that of the non-neighboring region (p=6.55×10$^{-15}$, Kolmogorov-Smirnov test) and the non-island cells (p=4.44×10$^{-16}$, Kolmogorov-Smirnov test).

DOI: https://doi.org/10.7554/eLife.45559.022

The following source data is available for figure 9:

**Source data 1.** Source data for the membrane curvature.

DOI: https://doi.org/10.7554/eLife.45559.023

inducing dynamic cell membrane protrusions. This phenomenon may be explained as follows; disruption of the cortical cytoskeleton leads to the loss of its elasticity, which had balanced the surface tension of the Par-island. The resulting imbalance in surface tension may cause the Par-island to shrink into a bowl or sphere shape, thereby bending the cell membrane outward and conferring protrusive activity to the cell membrane. In contrast, when membrane affinity is quite low, as in the case of Par3ΔMAR, Par-island shape is not affected by either cortical cytoskeleton elasticity or membrane

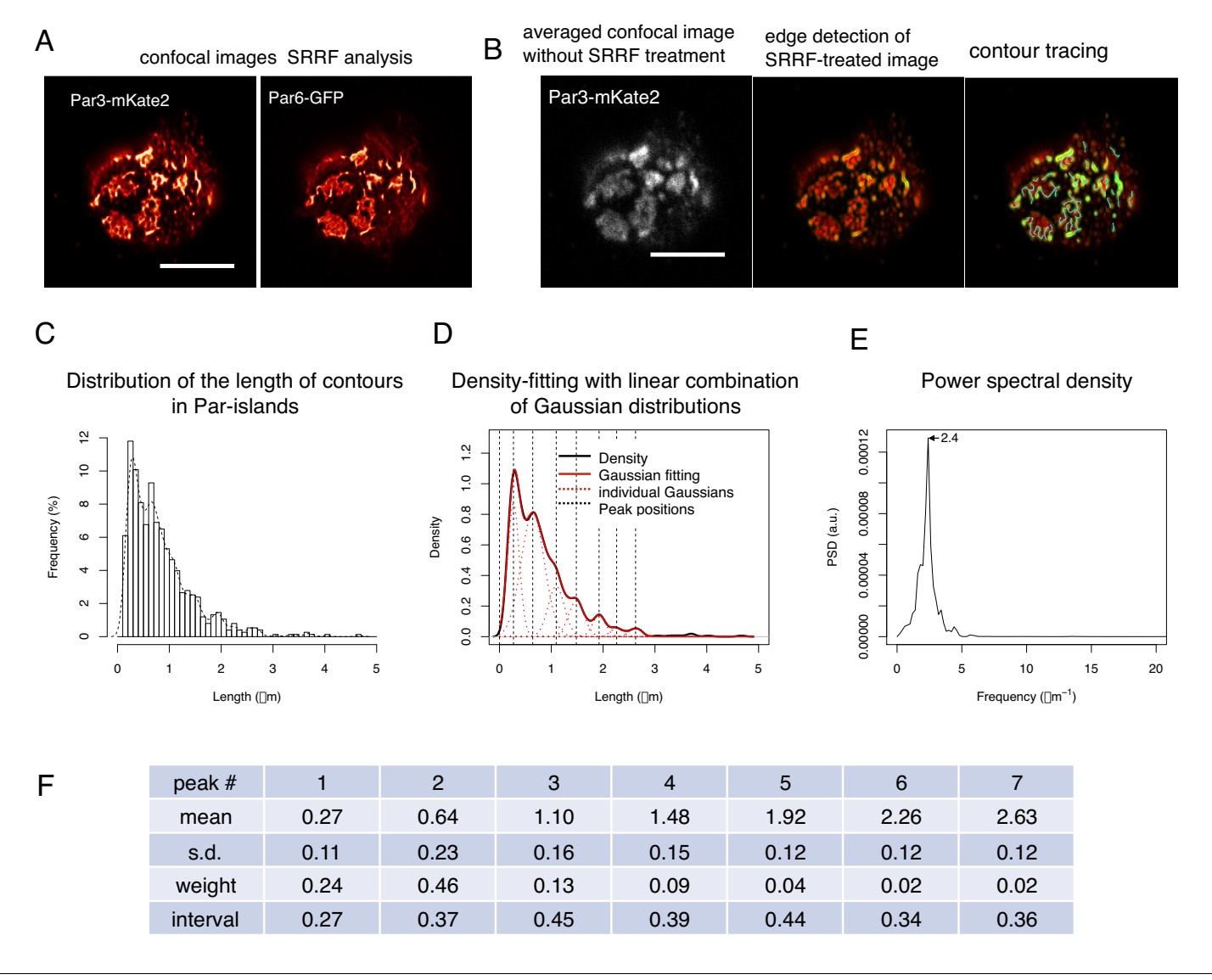

**Figure 10.** SRRF-processed confocal images reveals a unit-like segment in Par-islands. (A) SRRF-processed confocal images of cells expressing both Par3-mKate2 and Par6-GFP. Scale bar, 5 μm. (B) Left. the average of 200 confocal images of Par3-mKate2 distribution in a cell that expresses Par3-mKate2 and Par6-GFP. Those 200 images were used for SRRF analysis (*Figure 8A*). The middle Panel shows the SRRF-processed image (*Figure 8A*) that was processed with edge detection (see Materials and methods). By this process, the continuous contours become clearly visible. Edges in the image were visualized in green. The right panel shows tracing of continuous contours in the edge-detected image (middle) (light blue lines). Scale bar, 5 μm. (C) The histogram showing the distribution of the continuous contour line lengths in the right panel of *Figure 8B* and its density plot (dot line). (D) Gaussian fitting of the density plot (*Figure 8C*). The density plot of the histogram (*Figure 8C*) was fitted with 7 Gaussian curves via the least square method. The averaged mean of individual Gaussian curves was 0.38 ± 0.062 (s.d.) μm for 754 contours from 28 cells. (E) Power spectral density for the second derivative of the contour distribution plot shown in *Figure 8C*. The major frequency was 2.4 μm$^{-1}$. (F) The list of means and s.d. of the 7 Gaussian curves, whose combination best fitted the density plot of the continuous contour lengths distribution shown in *Figure 8C,D* (see Materials and methods).

DOI: https://doi.org/10.7554/eLife.45559.024

The following source data is available for figure 10:

**Source data 1.** Source data for the contour line length of SRRF images.

DOI: https://doi.org/10.7554/eLife.45559.025

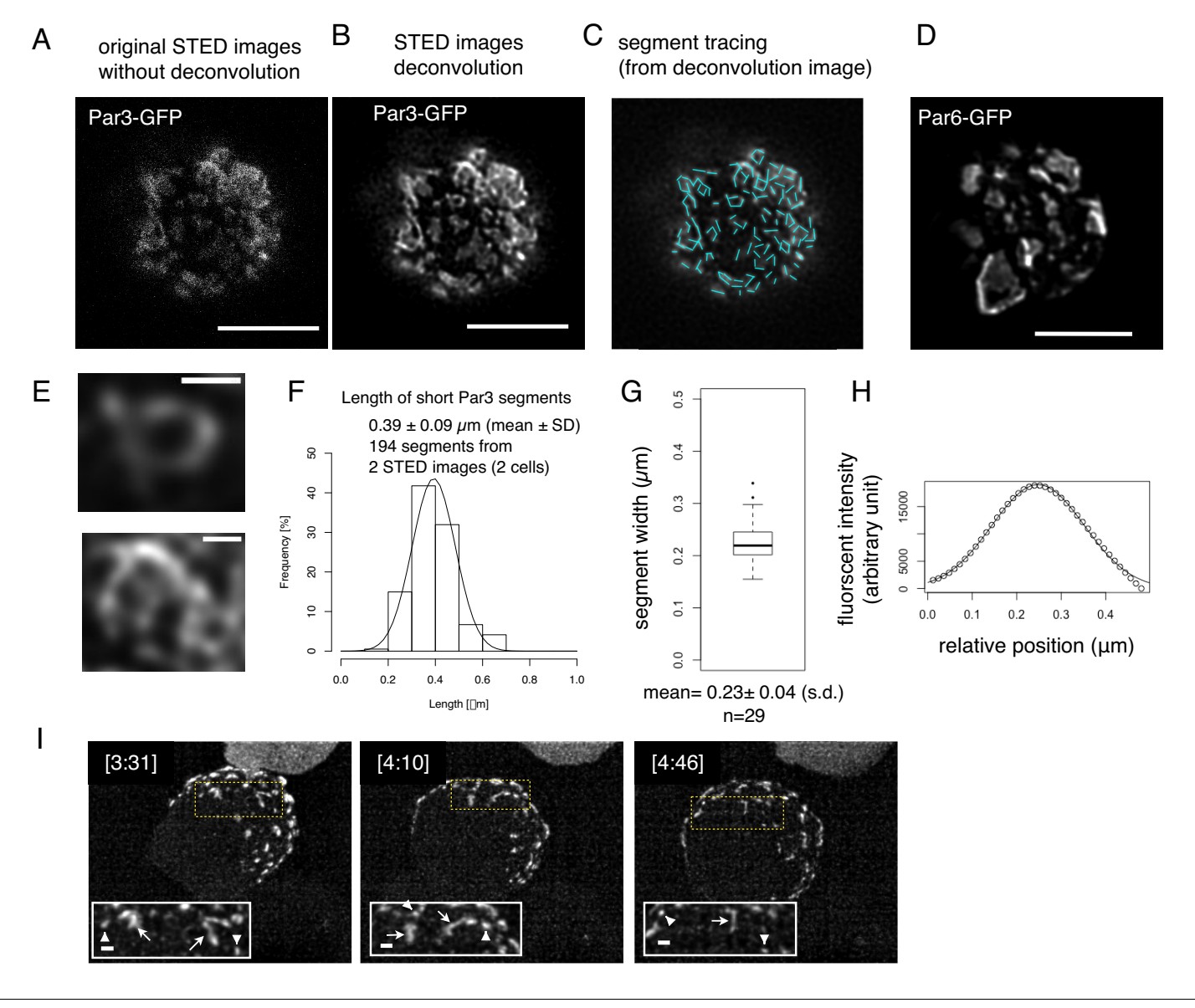

**Figure 11.** Structual analysis of the assembly state of the Par-complex using STED microscopy. (**A**) STED image of a cell that expresses Par3-GFP. The distribution of GFP was detected via indirect immunofluorescence staining. Scale bar, 5 μm for (**A–D**). (**B**) Deconvoluted image of the super-resolution (STED) image (**A**). (**C**) Tracing of the segments is indicated as light blue straight lines in the deconvoluted STED image (**B**). See Materials and methods for details. (**D**) Deconvoluted STED images of cells that expressed Par6-GFP together with pMT-myc-Par3, followed by GFP immunostaining. (**E**) Magnified views of the cell in (**B**) visualize the meshwork composed of unit-like segments. Scale bar, 0.5 μm. (**F**) Distribution of the length of individual segments constructing Par-islands visualized by Par3-GFP and its Gaussian fitting. See *Figure 8* for measurements. The mean value of the single segment lengths was 0.39 ± 0.09 (s.d.) μm based on 194 segments from 2 STED images for two cells including (**A**). (**G**) Distribution of half widths of segments composing Par-islands that were visualized by GFP staining in the two cells containing the cell shown in (**A**). The mean is 0.23 ± 0.04 μm (n = 29). (**H**) An example of Gaussian fitting of the fluorescence intensity distribution across the segment width visualized by immunofluorescence-staining for GFP. The mean half width = 0.23 ± 0.04 (s.d.) μm. (**I**) Rod and string structures of the Par3-mKate2 appearing in 3D time-lapse images of a cell expressing Par3-mKate2 during the period of Par-dot formation and development. Four time points were selected from *Video 2*. Scale bar, 2 μm. Insets display the magnification of a part of the image. Arrowheads indicate Par-dots, arrows, rods, and strings. Scale bar, 0.5 μm.

DOI: https://doi.org/10.7554/eLife.45559.026

The following source data is available for figure 11:

**Source data 1.** Source data for the length of segment of STED images.

DOI: https://doi.org/10.7554/eLife.45559.027

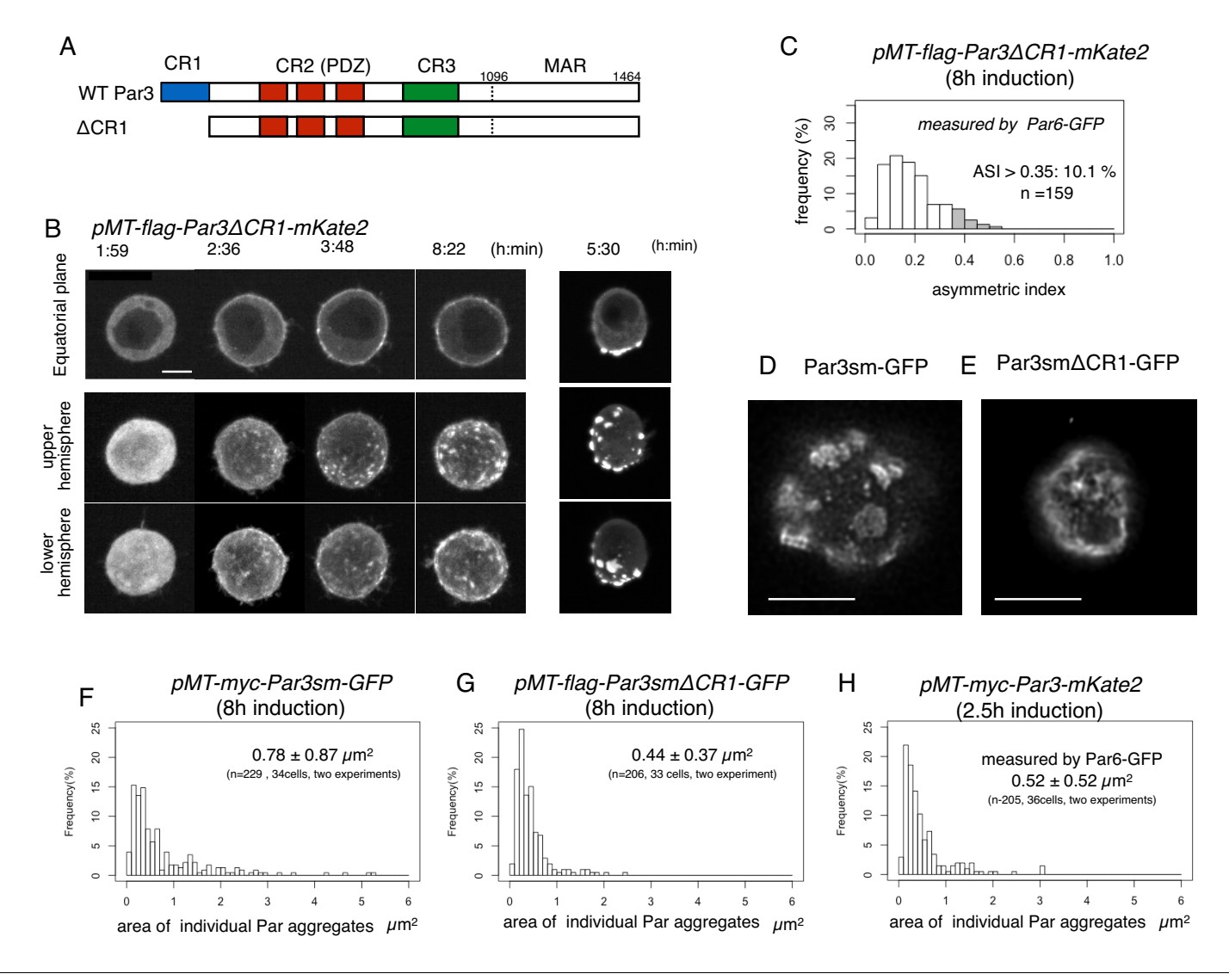

**Figure 12.** CR1 domain of Par3 is required for the formation of large regular meshwork of Par-islands. (**A**) Schematic representation of the functional domains of Par3 and ΔCR1 mutant constructs. (**B**) Time-lapse imaging of the distribution of Par6-GFP in cells inducing the expression of Par3ΔCR1-mKate2 under the same condition for Par3WT-mKate2. The right column is the same as that of *Figure 4A* for comparison with the case for Par3WT-mKate2 expression. Time is indicated in hr:min after CuSO₄ addition. In (**B**), images at the equatorial plane (top panels), and stacked images of the upper hemisphere (middle panels), and the lower hemisphere (bottom panels) are shown. Scale bar, 5 μm. (**C**) The distribution of ASI for cells that have induced Par3ΔCR1-mKate2 (Left, the mean value = 0.19 ± 0.10 for n = 159 cells). The gray part of histograms indicates the fraction of cells having ASI in the range out of the ASI distribution for memGFP-expressing cells (ASI > 0.35, see *Figure 3B*, and *Figure 7A*). Cells in this range are 10.1% of cells with cortical Par3ΔCR1-mKate2 (mean ASI value = 0.40 ± 0.05). This polarized cell population (ASI > 0.35) is significantly different (p=1.1×10⁻⁸, Fisher's exact test with post-hoc Bonferroni correction) from that for wild type myc-Par3-mKate2 (*Figure 7A*). Quantification was performed 8 hr after CuSO₄ addition. (**D and E**) The deconvolved STED images of the cells that expressed Par3smWT-GFP (**D**) and Par3smΔCR1-GFP (**E**). In both cells, the endogenous Par3 was knocked down by dsRNA. The distribution of GFP was detected by indirect immunofluorescence staining. Scale bar, 5 μm. (**F–H**) The distribution of the size of Par aggregates in cells that have induced (**F**) flag-Par3sm-GFP (0.78 ± 0.87 μm² for 34 cells), or (**G**) flag-Par3smΔCR1-GFP (0.44 ± 0.37 μm² for 33 cells). Quantification was performed 8 hr after CuSO₄ addition onwards. The endogenous Par3 had been knocked down by dsRNA. The size of Par aggregates in Par3ΔCR1-expressing cells is significantly smaller than that of Par3-expressing cells (p=0.000207, Kolmogorow-Smirnov test). (**H**). The distribution of the size of Par aggregates in cells expressing myc-Par3-mKate2 (0.52 ± 0.52 μm² for 36 cells) at 2.5 hr after CuSO₄ addition onwards, where the majority of Par aggregates are at the dotty state. The distribution is not significantly different from (**G**) (p=0.63, Kolmogorow-Smirnov test). In (**F–H**) CuSO₄ was added at 2 days post-transfection.

DOI: https://doi.org/10.7554/eLife.45559.028

The following source data and figure supplements are available for figure 12:

**Source data 1.** Source data for the ASI and island size of Par3ΔCR1 mutant.

*Figure 12 continued on next page*

*Figure 12 continued*

DOI: https://doi.org/10.7554/eLife.45559.032

**Figure supplement 1.** The suppression of the endogenous Par3 by RNAi.

DOI: https://doi.org/10.7554/eLife.45559.029

**Figure supplement 2.** Expression level of myc-Par3sm-GFP and flag-Par3ΔCR1sm-GFP, and effect of RNAi.

DOI: https://doi.org/10.7554/eLife.45559.030

**Figure supplement 2—source data 1.** Source data for the Western blotting image.

DOI: https://doi.org/10.7554/eLife.45559.031

affinity, and its shape would be determined only by the surface tension of Par3-islands. Under these conditions, we found that the Par-complex forms small cytoplasmic droplets, which subsequently coalesce into a spherical, densely packed structure, suggesting that phase separation takes place between the Par-complex aggregates and the cytoplasm (*Hyman et al., 2014*). A similar cytoplasmic droplet formation for a C-terminal truncated form of Par3 has been observed in *Drosophila* oocytes (*Benton and St Johnston, 2003*).

## Molecular network of the Par-complex in the island state

In this study, we revealed that a Par-island is a meshwork of various polygonal shapes, which appear to be built up of unit-like segments with an average length of approximately 0.4 µm. Isolated fragments such as single fragments and structures made up of a few connected fragments were observed during the development of Par-islands via live-imaging. These observations suggested that these isolated fragments assembled into a meshwork to form islands. Par-islands change shape rapidly during their movement along the cortex, and sometimes fuse with small pieces and also release pieces of different sizes, raising the possibility that Par-islands and small free fragments are mutually exchangeable. The factors that determine the size of these unit segments need further investigation.

Par3 is known to polymerize in vitro via the CR1 domain at its N-terminus to form a helical polymer of 8-fold symmetry (*Zhang et al., 2013*; *Feng et al., 2007*). Whether Par3 polymers are involved in the cortical cluster of the Par complex remain unclear. Our super-resolution microscopic observations and the ability of Par3 to form filaments lead to the simple hypothesis that the unit segment of a Par-island is formed by Par3 polymers as the core structure. This hypothesis is compatible with our observation that the overexpression of Par3ΔCR1 no longer forms a regular meshwork of Par-complex aggregates. While there are many possibilities via which Par3 filaments may form a unit segment, a single Par3 polymer may form a single segment. Another possibility is for Par3 polymers to be aligned along the long axis of the segment. Since Par6 and aPKC bind the PDZ and CR3 domains of Par3, respectively, Par6 and aPKC can act as cross-linkers between segments (*Feng et al., 2007*). Given the phenotype of ParS980A overexpression, the association of Par3 and aPKC by aPKC phosphorylation may confer flexibility and dynamism to the structure and/or assembly of the segmental elements. These hypotheses need to be tested in future studies. A surprising finding is that Par3ΔCR1 still has an ability to form Par-complex aggregates while those aggregates show a small size with no regular meshwork structure. We speculate that even in the absence of the CR1 domain, Par6 and aPKC cross-link multiple Par3ΔCR1 to form small aggregates via their multiple binding sites on Par3.

## Dynamic behaviors of Par aggregation and mechanisms for clustering

We have reproducibly obtained the data showing no significant effect of Y27632 on the behavior of Par-islands. Because cortical flow involving Myosin II is a major driving force for *C. elegans* polarization (*Munro et al., 2004*; *Motegi and Sugimoto, 2006*), the dynamic behavior of Par-islands is not likely to depend on the typical Myosin II cortical flow. Moreover, we observed no directional movement of Par-dots or Par-islands during the process of asymmetric clustering nor at the steady state. These two observations suggest no significant involvement of cortical flow in Par-island dynamics in the S2 cell reconstruction system, while we do not exclude the possibility that Y27632-insensitive Myosin might work for the dynamics of Par-islands even though it is unlikely to operate in a directional way.

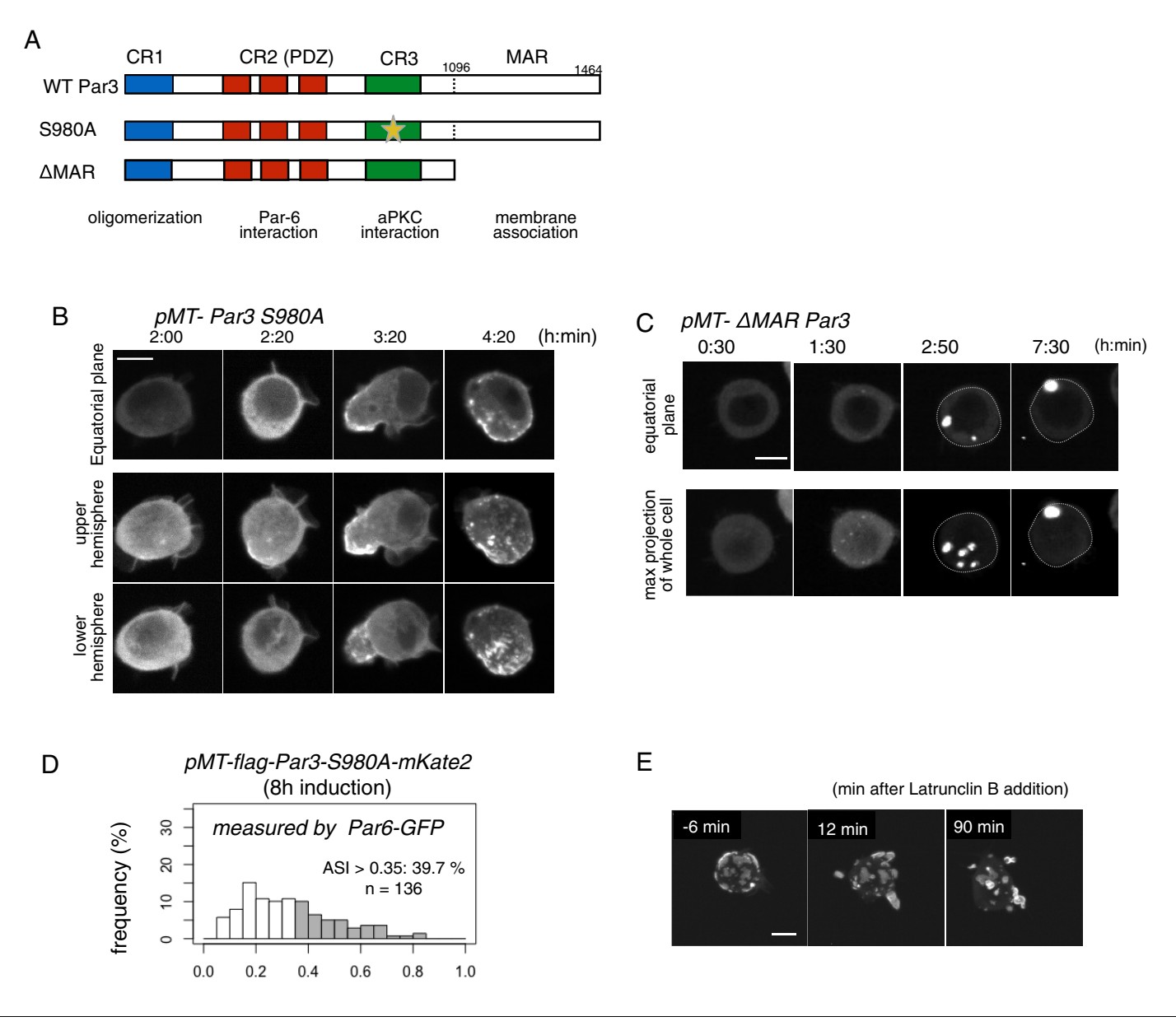

**Figure 13.** Role of the phosphorylation of Par3 and membrane binding. (A) Schematic representation of the functional domains of Par3 and S980A and ΔMAR mutant constructs used in this study. (B–C) Time-lapse imaging of the distribution of Par6-GFP in cells where Par3S980A-mKate2 (B) and Par3ΔMAR-mKate2 (C) were induced by the *Metallothionein* promoter. Time is indicated in h:min after CuSO4 addition. In (B), images at the equatorial plane (top panels), and stacked images of the upper hemisphere (middle panels), and the lower hemisphere (bottom panels) are shown. In (C), the lower panels show maximum-projection images of the whole cell. Scale bar, 5 μm. (D) The distribution of ASI is shown for cells that have induced Par3S980A-mKate2 (F, the mean value = 0.33 ± 0.18 for n = 139 cells). The gray part of histograms indicates the fraction of cells having ASI in the range out of the ASI distribution for memGFP-expressing cells (ASI > 0.35, see *Figure 3B*, and *Figure 6A*). Cells in this range are 39.7% for Par3S980A-mKate2 (mean ASI value = 0.52 ± 0.13). The polarized cell population (ASI > 0.35) is significantly altered (p=0.04486 for Par-3S980A-mKate2, Fisher's exact test with post-hoc Bonferroni correction) compared with that of wild-type myc-Par3-mKate2. Quantification was performed 8 hr after CuSO4 addition. In all images, CuSO4 was added at 2 days post-transfection. (G) Time-lapse imaging of the effects of actin disruption on the Par-islands. At 8 hr after Par3-mKate2 induction, cells were treated with Latrunculin B. Par-islands rapidly became round and/or promoted membrane protrusion. Faint fluorescent islands face the bottom of the dish. See *Video 6*. Scale bar, 5 μm.

DOI: https://doi.org/10.7554/eLife.45559.033

The following source data is available for figure 13:

**Source data 1.** Source data for the ASI of Par3S980A mutant.

DOI: https://doi.org/10.7554/eLife.45559.034

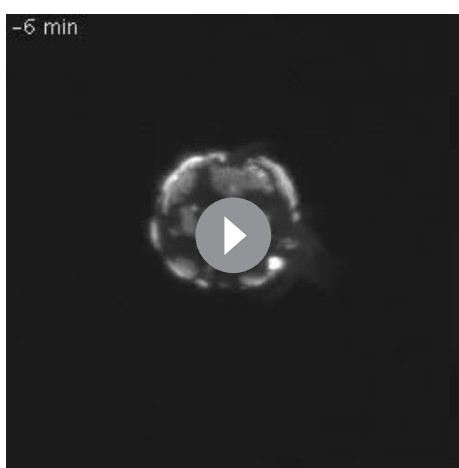

**Video 6.** 3D time-lapse movie of a S2 cell monitored by Par6-GFP, 3 min after the addition of Latrunculin B, at 8 hr induction of myc-Par3-mKate2, following 2 days transfection of pMT-myc-Par3-mKate2, pAct-Par6-GFP, and pAct-aPKC.

DOI: https://doi.org/10.7554/eLife.45559.035

How can Par islands cluster asymmetrically and dynamically behave? A possible mechanism underlying Par-island dynamics is diffusion and/or tension. We think that the situation where advection such as cortical flow surpasses diffusion for transportation is the following two cases. Firstly, the scale of the system is large enough. *C. elegans* one-cell embryo and *Drosophila* oocytes are good examples. In these cases, the random diffusion and capture mechanism without advection will take a much longer time (*Goehring et al., 2011*). While *C. elegans* one-cell embryo is approximately 50 μm long along the anterior-posterior axis, the size of S2 cells is approximately 10 μm in diameter (*Figure 9B*), which seems to be nearly marginal about which of advection and diffusion is efficient to polarize cells. The mutual capture of islands might work well to cluster islands in one pole.

Secondly, the time scale of clustering in S2 cell system is an order of hours, while the polarization of *C. elegans* one-cell embryo or of *Drosophila* neuroblasts occurs in an order of minutes (within 15 min, and 5 min, respectively). Such a big difference in the time scale of Par-island polarization, together with its non-directional movement, is likely explained by the notion that polarized clustering of Par-islands occurs through diffusion and mutual capture that might be driven by phase separation with surrounding cortex.

In conclusion, while we do not exclude the involvement of myosin motor in Par island dynamics, it is more likely to be driven by diffusion mechanism on the two-dimensional cortex.

## The two states of the Par-island distribution at steady state

An interesting property of Par-islands is that they are not unified into one large island under the cell membrane, even when polarized. Overexpression of Par3ΔMAR or Par3S980A is an exception. In the latter case, rapid and enhanced formation of the cortical Par-complex does not seem to initially permit separate island formation, and a large, transient dome is formed instead. In the former case, the Par-complex aggregates to form one large sphere. This cytoplasmic phenomenon is likely to be due to a phase separation between the Par-complex and the cytoplasm. Considering this property of the Par-complex, the unique feature of Par-islands associated with the cell membrane may reflect phase separation in two dimensions.

Steady-state Par-island distribution in a cell may be classified into two different states, polarized and non-polarized. While we failed to identify a single parameter correlating these two states (*Figure 5A,B*), our analysis shows that the two states of island distribution are nearly fixed during the formation of islands (*Figure 6A*). Because the position of island formation appears to be stochastic, variation in the position of Par-island formation across the cell may explain the two localization patterns of Par-islands. Since Lgl distribution is largely complementary to dots and islands, this molecule may contribute to stabilize the two states of island distribution at the cellular scale (*Betschinger et al., 2003*; *Guo and Kemphues, 1995*). Thus, these two different states of Par-island distribution may be the outcome of two stable solutions of the reaction diffusion system (*Chau et al., 2012*; *Goehring et al., 2011*), where a negative regulator Lgl is involved (*Betschinger et al., 2003*). The initial condition, which is possibly determined by a stochastic distribution of islands, may select one of the two stable patterns in a cell. We propose that such cell-scale patterning is coupled with local phase separation of Par-islands as previously described for the membrane lipid domain (*John and Bär, 2005*).

In summary, we have developed a potential Par complex-polarization system upon induction of Par3 in non-polar S2 cells, which provides a useful model for cell-autonomous cell polarization, allowing us to easily manipulate gene expression and image at the super-resolution level. One intriguing

challenge will be the coupling of mitosis with cell polarization in this system to induce asymmetric division.

# Materials and methods

## Key resources table

| Reagent type (species) or resource | Designation | Source or reference | Identifiers | Additional information |
|---|---|---|---|---|
| Genetic reagent (*D. melanogaster*) | y, w, Baz-GFP | Bloomington Drosophila Stock Center | BDSC:51572; FLYB:FBtc0099880 | FlyBase symbol: P{PTT-GC}baz CC01941 |
| Cell line (*D. melanogaster*) | S2 | Invitrogen, Thermo Fisher Scientific | Invitrogen:R69007 | |
| Antibody | Rabbit polyclonal anti-aPKC | Santa Cruz | Santa_Cruz: sc-216 | IHC (1:1000) WB (1:3000) |
| Antibody | Mouse monoclonal anti-Par3 | *Ohshiro et al. (2000)* | | IHC (1:100) |
| Antibody | Rabbit polyclonal anti-Par3 | *Ohshiro et al. (2000)* | | IHC (1:1000) WB (1:3000) |
| Antibody | Rabbit polyclonal anti-Par6 | *Izumi et al. (2004)* | | IHC (1:1000) |
| Antibody | Chicken polyclonal anti-Myc | Bethyl | Bethyl:A190-103A | IHC (1:1000) |
| Antibody | Mouse monoclonal anti-Miranda | *Ohshiro et al. (2000)* | | IHC (1:100) |
| Antibody | Rabbit polyclonal anti-Lgl | *Ohshiro et al. (2000)* | | IHC (1:1000) WB (1:1000) |
| Antibody | Guinea pig polyclonal anti-Cdc42 | Ulrich Tepass (University of Tronto) | | IHC (1:100) |
| Antibody | Chicken polyclonal anti-GFP | aves | aves:GFP-1020 | IHC (1:1000) |
| Antibody | Mouse monoclonal anti-GFP | chemicon | chemicon: MAB3580 | IHC (1:1000) |
| Antibody | Rabbit polyclonal anti-GFP | MBL | MBL:598 | IHC for Larva brain (1:1000) WB (1:2000) |
| Antibody | Mouse monoclonal anti-Flag | Sigma | Sigma:F3165 | IHC (1:1000) |
| Antibody | Alexa 488, Cyanin3, Cyanin5 secondaries | Jackson Immuno Research | | IHC for S2 cells (1:4000) IHC for Larva brain (1:200) |
| Antibody | anti-alpha tubulin | sigma | Sigma:T9026 | WB (1:15000) |
| Chemical compound, drug | Latrunclin B | Wako | Wako:129–05101 | final concentration, 1 $\mu$M |

*Continued on next page*

*Continued*

| Reagent type (species) or resource | Designation | Source or reference | Identifiers | Additional information |
|---|---|---|---|---|
| Chemical compound, drug | CuSO₄ | Wako | Wako:034–04445 | final concentration, 1 mM |
| Recombinant DNA reagent | pMT/V5-His B (Gateway vector) | Invitrogen, Thermo Fisher Scientific | Invitrogen:V412020 | |
| Recombinant DNA reagent | pAc5.1/V5-His B (Gateway vector) | Invitrogen, Thermo Fisher Scientific | Invitrogen: V411020 | |
| Recombinant DNA reagent | pAc-Gal4 (Gateway vector) | Addgene | Addgene:24344 | provided from Liqun Luo (Stanford University) |
| Recombinant DNA reagent | pDAMCS (Gateway vector) | this paper | | Progenitors: pAct5C0 plasmids (BglII-XhoI fragment); Gateway vector pUC19 |
| Recombinant DNA reagent | pUAST | *Brand and Perrimon (1993)* | | |
| Recombinant DNA reagent | Par6-GFP (plasmid) | this paper | | Progenitors: Par6 and GFP (cDNA); Gateway vector pAc5.1/ V5-His B |
| Recombinant DNA reagent | aPKC-GFP (plasmid) | this paper | | Progenitors: aPKC and GFP (cDNA); Gateway vector pAc5.1/ V5-His B |
| Recombinant DNA reagent | myc-Par3-mKate2 (plasmid) | this paper | | Progenitors: Par3 and mKate2 (cDNA); Gateway vector pMT/V5-His B, pAc-Gal4 or pDAMCS |
| Recombinant DNA reagent | myc-Par3-GFP (plasmid) | this paper | | Progenitors: Par3 and GFP (cDNA); Gateway vector pMT/V5-His B, pAc-Gal4 or pDAMCS |
| Recombinant DNA reagent | membrane-GFP | this paper | | Progenitors: myristoylation tag from Fyn and GFP (cDNA); Gateway vector pAc5.1/ V5-His B |
| Software, algorithm | Huygens Professional | Scientific Volume Imaging | | ver. 17.10 |
| Other | DAPI | nacalai tesque | nacalai_tesque: 11034–56 | IHC for S2 cells (1:4000) IHC for Larva brain (1:200) |

## Cell culture

S2 cell (Invitrogen:R69007) culture and transfection were performed as previously described (*Ogawa et al., 2009*). Expression vectors were transfected at 2 days prior to microscopic or western

blot analysis. For induction of the *Metallothionein* promoter, 100 mM CuSO$_4$ solution was added to a medium at a final concentration of 1 mM.

## Live cell imaging

Cells were mounted on a 35-mm glass-bottom dish coated with 15 µg/ml poly-L-ornithine and incubated at 25℃ for 30 min, followed by microscopic analysis. Images were taken at a 1 µm z-interval with a spinning disk confocal microscopy CSU-W1 (Yokogawa, Tokyo, Japan) equipped with a sCMOS camera Neo (Andor, Belfast, Northern Ireland) and MetaMorph software (Molecular Devices, San Jose, CA).

## Immunostaining

For immunostaining of S2 cells, transfected cells were mounted on a poly-L-ornithine-coated cover slip and fixed with 4% paraformaldehyde in PBS for 15 min at room temperature. Cells were washed with PBS, followed by treatment with 0.1% Triton-X100 in PBS for 15 min. After washed with PBS, cells were treated with a blocking buffer containing 2% BSA in PBS for 30 min and incubated with primary antibodies in the blocking buffer for 30 min, followed by incubation with secondary antibodies for 30 min. Immunostained cells were embedded in mounting medium PermaFluor (Thermo Fisher Scientific) and analyzed with a confocal microscopes LSM510 (Zeiss, Oberkochen, Germany). For super-resolution microscopy, samples were embedded in ProLong Glass Andifade Mountant (Thermo Fisher Scientific) and analyzed with a comfocal microscopes LSM880 (Zeiss, Oberkochen, Germany).

For immunostaining of *Drosophila* neuroblasts, brains isolated from third instar larvae were fixed with 4% paraformaldehyde in PBS for 20 min at room temperature. Samples were treated with the blocking buffer for 1 hr, followed by incubation with primary antibodies and secondary antibodies for overnight and 2 hr, respectively. Samples were then embedded in Vectashield H-1000 (Vector Laboratories) and analyzed with a confocal microscope FV1000 (Olympus, Tokyo, Japan).

Primary antibodies used were anti-aPKC (rabbit polyclonal, used at 1:1000, Santa Cruz), anti-Par-3 (rabbit polyclonal, used at 1:1000, or mouse monoclonal, used at 1:100) (*Ohshiro et al., 2000*), anti-Par-6 (rabbit polyclonal, used at 1:1000) (*Izumi et al., 2004*), anti-Myc (chicken polyclonal, used at 1:1000, Bethyl), anti-Flag (mouse monoclonal, used at 1:1000), anti-Miranda (mouse monoclonal, used at 1:100) (*Ohshiro et al., 2000*), anti-Lgl (rabbit polyclonal, used at 1:1000) (*Ohshiro et al., 2000*), anti-GFP (chicken polyclonal, Betyl, mouse monoclonal, Chemicon, and rabbit polyclonal, MBL, used at 1:1000), anti-Cdc42 (guinea pig polyclonal, used at 1:100, a kind gift from U. Tepass, University of Toronto, Canada). Secondary antibodies used were anti-Rabbit Alexa Fluor488 (donkey polyclonal, Jackson Immuno Research), anti-mouse Alexa Fluor488 (donkey polyclonal, Jackson Immuno Research), anti-chicken Alexa Fluor488 (donkey polyclonal, Jackson Immuno Research), anti-rabbit Cyanin3 (Donkey polyclonal, Jackson Immuno Research), anti-mouse Cyanin3 (donkey polyclonal, Jackson Immuno Research), anti-chicken Cyanin3 (donkey polyclonal, Jackson Immuno Research), anti-rabbit Alexa Fluor647 (donkey polyclonal, Jackson Immuno Research), anti-guinea pig Cyanin5 (donkey polyclonal, Jackson Immuno Research). All secondary antibodies were used at 1:4000 or 1:200 for staining of S2 cells or *Drosophila* 3$^{rd}$ instar larvae, respectively.

## Super-resolution microscopy

For the super-resolution radial fluctuations (SRRF) method, S2-cells were transfected with *pMT-myc-Par3-mKate2*, *pAct-Par6-GFP* and *pAct-aPKC*. For SRRF method, confocal imaging was performed using LSM880 (Zeiss) with an objective lens Plan-Apochromat 63x/1.4 Oil DIC M27 (Zeiss). A series of 200 frames was obtained for each cell with a pixel size of 53 nm and 160 ms exposure time. Drift-correction and reconstruction of SRRF images were performed with an ImageJ plug-in NanoJ-SRRF (*Gustafsson et al., 2016*).

Using SRRF-processed images, Par3 contour lengths along the meshwork were manually traced with Fiji. Each image was overlaid by an edge-enhanced image generated with the Sobel filter, to highlight Par3 contour shapes. Lengths between their terminal ends and/or branching points were measured. A histogram and a density plot were generated from all contour lengths, and the shape of the density plot was fitted with a linear combination of 7 Gaussian curves by a fitting function

implemented in R with the non-linear least square method. Power spectral density of the second derivative of the density plot was calculated using fast Fourier transform method with R.

For Stimulated emission depletion (STED) imaging, S2-cells were transfected with *pMT-myc-Par3-GFP*, *pAct-Par6* and *pAct-aPKC*, or *pMT-myc-Par3*, *pAct-Par6-GFP* and pAct-aPKC, followed by CuSO$_4$ addition for induction 2 days following transfection. Cells were fixed for immune-staining for GFP at 8 hr post-induction. STED imaging was performed using TCS SP8 STED 3X microscope (Leica, Wetzlar, Germany) with an objective lens HC PL APO 93X/1.30 GLYC (Leica). Deconvolution was performed with a deconvolution software package Huygens Professional (version 17.10, Scientific Volume Imaging, Hilversum, Netherlands).

Deconvoluted STED images were used for the analyses of Par3 segment lengths and widths. The segment length was defined as a shortest length between terminal ends, corners and/or branching points of Par3 contours, and manually traced with Fiji. The segment width was given by the full width at half maximum (FWHM) of a Gaussian-fitted signal distribution orthogonal to each Par3 segment.

## Quantification of asymmetry and statistics

The equatorial z-plane of each cell was analyzed for the estimation of asymmetric index (ASI) (see also *Figure 3A*). The cell perimeter was traced by a 0.5 µm-width line and the signal intensity along the line was measured with Fiji. The signal intensities were summed up along the half (L) of the total perimeter length (2L). The difference between this value and that of the other half was calculated and normalized by the total signal intensity along the perimeter. This measurement was done starting from every pixel along the perimeter (one pixel = 0.108 µm), The maximum value of them was defined as ASI. Cell with an ASI larger than 0.35 was defined as polarized cell. Each experiment to measure the distribution of ASI was independently duplicated, and the results of the two experiments were combined to make a single histogram for ASI distribution. The statistical significance of polarized cell population was analyzed by Fisher's exact test with post-hoc Bonferroni correction for multiple comparisons (*Figures 3A,* and *10E,F*). Statistical analyses were performed with R software.

## Quantification of membrane curvature

The equatorial z-plane of each cell was analyzed for the estimation of membrane curvature radius. The cell membrane was defined by membrane-GFP signal. A Par-island region was defined as a continuous compartment of the cell membrane that co-localized with continuous myc-Par3-mKate2. A non-neighboring region was defined in cells with Par-islands as a continuous membrane compartment of less than 5 µm at a distance of more than 5 µm from Par-island regions. A membrane region in non-island cells was defined as a continuous membrane compartment of less than 5 µm in cells without Par-islands. For non-island cells, 5–6 regions were analyzed per cell. Coordinates of two ends and a mid-point of each region were taken. The circle that passes through the three points was solved from the coordinates, and its radius was shown as a curvature radius in *Figure 9*. Measurements was performed with Fiji. Calculations and statistics were performed with R.

## Quantification of the size of Par aggregates

Bottom surface images of cells were analyzed to measure the size of Par aggregates. Region of interests (ROI) were made along the perimeters of Par aggregates by tracing manually with Fiji. Aggregates that contacted with a limit of an optical section were excluded from the analysis. The area of each ROI was measured, and shown as the size of Par aggregates in *Figure 12*.

## Western blot analysis

Whole cell extracts of the untransfected S2 cells and the transfected S2 cells were subjected to SDS-polyacrylamide gel electrophoresis. Primary antibodies used were anti-Par3 antibody (rabbit polyclonal, used at 1:1000), anti-alpha-tubulin (rat monoclonal, Santa Cruz). Secondary antibodies used were horseradish peroxidase (HRP)-conjugated anti-mouse antibody (sheep polyclonal, used at 1:3000, GE Healthcare), HRP-conjugated anti-rabbit antibody (sheep polyclonal, used at 1:3000, GE Healthcare) and HRP-conjugated anti-chicken antibody (donkey polyclonal, used at 1:250, SA1-300, Thermo Fisher Scientific). Protein level was analyzed by chemiluminescence with Chemi-Lumi One L (Nacalai tesque, Kyoto, Japan) and quantified with an image analyzer LAS-3000 system (Fujifilm, Tokyo, Japan). To compare the expression level of the overexpressed fluorescent protein per cell

between two different transfectants or with the endogenous Par3 proteins, transfection efficiency for each sample was calculated by counting fluorescence-positive cells and negative cells. The ratio of the expression level per cell was calculated by dividing the measured staining intensity on the western blot by the transfection efficiency.

## Knock-down experiment

Long double-stranded RNAs (dsRNAs) were used for RNA interference (RNAi) in S2 cells as previously described (*Bettencourt-Dias and Goshima, 2009*). For knocking-down *Par6* or *aPKC*, dsRNA was synthesized with MEGAscript T7 Transcription Kit (Ambion, Thermo Fisher Scientific) according to the manufacturer's instructions, by using *pBS-T7/Par-6/T7* or *pBS-T7/aPKC/T7* plasmid directly as a template, which contains the full-length *Par6* or *aPKC* ORF flanked by two T7 promoters, respectively. For knocking-down *Lgl* or *Par3*, dsRNA was synthesized by using a PCR amplicon consisting of the full-length *Lgl* ORF or a part of the *Par3* ORF (906–1226 nucleotides) flanked by T7 promoters as a template. PCR was performed by using *pUAS-Flag-Lgl* or *pUAS-myc-Par3* as a template and primers shown in below:

*Lgl*, 5'-TAATACGACTCACTATAGGGATGGCAATAGGGACGCAAACAGGGGCTTTAAAAGTT-3' and 5'-TAATACGACTCACTATAGGGTTAAAATTGGCTTTCTTCAGGCGCTGTTTTTGGCGTTCCAA-3'; *Par3*, 5'- TAATACGACTCACTATAGGGTGAATCCATCAGGGAGAAGG-3' and 5'- TAATACGACTCACTATAGGGCTCGGCCACCTTAGAGTCAC-3'.

The dsRNA sequence for knocking-down Par3 was selected according to an RNAi database DRSC (DRSC25558) (*Mohr et al., 2015*).

dsRNAs were added to the culture media at a final concentration of 4.5 µg/ml at 2–3 hr following transfection of expression plasmids.

## Plasmid construction

To construct expression vectors under control of an *actin (act5c)* promoter, *Drosophila Par6*, *Par3* or *aPKC* ORF, or *Par6*, *aPKC* or *Fyn* myristoylation tag (5'- ATGGGCTGTGTGCAATGTAAGGATAAAGAAGCAACAAAACTGACG-3') conjugated with *GFP* at the C-terminus (*Par-6-GFP*, *aPKC-GFP*, *membrane-GFP*) was inserted into *pAc5.1/V5-His B* plasmid (Invitrogen, Thermo Fisher Scientific).

To construct an expression vector for *Par3* under control of the *Gal4-UAS* system, *Par3* conjugated with *Myc* or *Flag* epitope and *mKate2* at the N-terminus and C-terminus, respectively, or *Lgl^3A* conjugated with *Flag* epitope was inserted into *pUAST* plasmid (*Brand and Perrimon, 1993*), and *Drosophila gal4* ORF, which was subcloned from *pAC-GAL4* plasmid, was inserted into *pDAMCS* plasmid. To construct *pDAMCS* expression plasmid, *BglII-XhoI* fragment of *pAct5C0* plasmid (*Thummel et al., 1988*) containing actin 5C promoter and poly(A) addition signals (and a small region of *hsp70* promoter) was cloned between *BamHI* and *SalI* site of *pUC19* plasmid. Then, a synthetic double-stranded oligonucleotide containing a multiple cloning site was inserted into the *BamHI* site between the actin 5C promoter and the poly(A) addition signals.

To construct expression vectors for *Par3* under control of the induction system, *Par3* conjugated with *Myc* epitope and mKate2 or GFP at the N-terminus and C-terminus, respectively, was inserted into *pMT* plasmid (Invitrogen, Thermo Fisher Scientific).

To construct expression vectors for RNAi-resistant form of *Par3* (*Par3sm*), silent mutations that do not alter amino acids were introduced into the dsRNA-targeted region in the *Par3* ORF as below, in which mutated nucleotides were underlined, 5'-TGAA<u>A</u>GCAT<u>T</u>AGAGA<u>A</u>AA<u>A</u>G AC<u>G</u>GC<u>G</u>GA<u>G</u>ATG TTG<u>T</u>TGAT<u>T</u>AT<u>A</u>AA<u>T</u>GA<u>A</u>TA<u>T</u>GGA<u>T</u>CG<u>C</u>CC<u>C</u>TTGGG<u>C</u>TT<u>G</u>AC<u>C</u>GC<u>C</u>CTGCCC<u>G</u>AT<u>A</u>AA<u>G</u>GA<u>A</u>CACG-GA<u>G</u>GC<u>G</u>G<u>T</u>TT<u>G</u>CTGGT<u>C</u>CAGCACGT<u>C</u>GA<u>A</u>CCC<u>G</u>G<u>A</u>AGT<u>C</u>GC<u>G</u>C<u>A</u>GA<u>A</u>AGG<u>G</u>GG<u>C</u>CGC TT<u>G</u>CGCCGCGAC<u>G</u>AC<u>C</u>GC<u>A</u>TT<u>C</u>T<u>C</u>GA<u>A</u>ATT<u>A</u>AC<u>G</u>GCATT<u>A</u>AA<u>A</u>TT<u>G</u>AT<u>C</u>GGC<u>C</u>T<u>G</u>AC<u>G</u>GA<u>G</u>AGC-CA<u>A</u>GT<u>C</u>CAGGA<u>A</u>CA<u>G</u>TTGCG<u>C</u>CG<u>C</u>GCC<u>C</u>TGGAG<u>T</u>CCAGC<u>G</u>A<u>A</u>CTGAG<u>G</u>GT<u>G</u>C<u>G</u>GG<u>T</u>CC TC<u>C</u>GCGG<u>C</u>GAC<u>C</u>GCAA<u>C</u>CG<u>C</u>CG<u>C</u>CAGCA<u>A</u>CGCGATT<u>C</u>C<u>A</u>AA<u>G</u>TCGCT<u>GA-3'</u>. A mutated DNA fragment (GeneArt, Thermo Fisher Scientific) was amplified by PCR and replaced with the corresponding region of *Par3WT-GFP* or *Par3ΔCR1-GFP* in *pMT* plasmid by using In-Fusion.

*pAC-GAL4* was a gift from Liqun Luo (Addgene plasmid # 24344; http://n2t.net/addgene:24344; RRID:Addgene_24344; *Potter et al., 2010*). *pUbq-Spd2-GFP* was a gift from Jordan Raff (University of Oxford, UK). Plasmids used for transfection were purified with Wizard Plus SV Minipreps DNA

Purification System (Promega, Madison, WI) or NucleoBond Xtra Midi (Macherey-Nagel, Düren, Germany).

## Acknowledgements

We thank Y Tsunekawa for technical advice, Y Otsuka for technical assistance, U Tepass for ant-Cdc42 antibody, J Raff for plasmids, T Nishimura for fly strains, S Hayashi, T Nishimura, and K Kawa-guchi, and S Yonemura for critical discussion. Some of the imaging experiments were performed at the RIKEN Kobe Light Microscopy Facility. IF is a recipient of RIKEN Special Postdoctoral Researcher Program.

## Additional information

### Funding

| Funder | Author |
|--------|--------|
| RIKEN | Fumio Matsuzaki<br>Ikumi Fujita<br>Kalyn Kono |

The funders had no role in study design, data collection and interpretation, or the decision to submit the work for publication.

### Author contributions

Kalyn Kono, Conceptualization, Data curation, Formal analysis, Validation, Investigation, Visualization, Methodology, Writing—original draft; Shigeki Yoshiura, Conceptualization, Supervision, Validation, Investigation; Ikumi Fujita, conceptualization, Data curation, Formal analysis, Validation, Investigation, Writing—review and editing; Yasushi Okada, Formal analysis, Investigation, Visualization, Research advice; Atsunori Shitamukai, Formal analysis, Validation, conceptualzation, Visualization, Methodology; Tatsuo Shibata, conceptualization, Validation, Writing—original draft, Writing—review and editing; Fumio Matsuzaki, Conceptualization, Supervision, Funding acquisition, Validation, Writing—original draft, Project administration, Writing—review and editing

### Author ORCIDs

Kalyn Kono (iD) https://orcid.org/0000-0002-1558-7153
Ikumi Fujita (iD) https://orcid.org/0000-0002-1161-0352
Yasushi Okada (iD) https://orcid.org/0000-0003-2601-3689
Atsunori Shitamukai (iD) https://orcid.org/0000-0003-4216-927X
Tatsuo Shibata (iD) http://orcid.org/0000-0002-9294-9998
Fumio Matsuzaki (iD) https://orcid.org/0000-0001-7902-4520

### Decision letter and Author response

Decision letter https://doi.org/10.7554/eLife.45559.038
Author response https://doi.org/10.7554/eLife.45559.039

## Additional files

### Supplementary files
• Transparent reporting form
DOI: https://doi.org/10.7554/eLife.45559.036

### Data availability
All data generated or analysed during this sturdy are included in the manuscript and supporting files. Source data files have been provided for all figures.

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
