## [Decision Letter]

Thank you for submitting your article "Reconstruction of Par-dependent polarity in apolar cells reveals a dynamic process of cortical polarization" for consideration by *eLife*. Your article has been reviewed by three peer reviewers, including Yukiko M Yamashita as the Reviewing Editor and Reviewer #1, and the evaluation has been overseen by Anna Akhmanova as the Senior Editor. The following individuals involved in review of your submission has also agreed to reveal their identity: Hongyan Wang (Reviewer #2).

The reviewers have discussed the reviews with one another and the Reviewing Editor has drafted this decision to help you prepare a revised submission.

Summary:

This manuscript by Matsuzaki and colleagues describes the induction of Par protein polarization in a normally non-polarized cell-type, *Drosophila* S2 cells, and characterizes structural aspects of Par complexes formed. They found that overexpression of Par3 in S2 cells is sufficient to induce polarization, where Par3's partners (aPKC and Par6) also polarize together. This polarization depends on Lgl, suggesting that this system recapitulates antagonistic system of cell polarization between Par complex and Lgl.

By utilizing temporal induction system, which provided a unique opportunity to study the process of polarization, they found that Par3 emerges as 'dots' on the cortex, and self-organizes into larger islands. By STED microscopy, they further show that Par3 'islands' likely have uniform/regular structure, indicating that it may be a self-assembled structure based on smaller units. In addition, they found that the functional domains of Par3 (CR1 and Serine 980 of CR3 domains) and the interactions between these domains contribute to the Par polarization in these polarity-induced S2 cells. Finally, actin cytoskeleton prevent the dynamic redistribution of islands into a spherical shape.

Overall, this is a very interesting study with novel findings on the dynamic assembly of Par-complex during polarization. This is a well-written paper with logical flow, making it easy to follow. This is an important contribution to the field, providing new understanding of how cell polarity may be established by Par3-Par6-aPKC system. The topic will be of great interest to readers of *eLife*.

Essential revisions that may require additional experiments:

1) The structures of the clusters formed with Par-3 or Par-3 lacking CR1 (the oligomerization domain) were not clearly different in the images shown. Perhaps the authors' super-resolution approach would reveal a clearer difference. This further assessment would be important for assessing whether the oligomerization domain has a role in forming filaments in cells. Depleting endogenous Par-3 may be needed for seeing a clear role of CR1.

2) Since the induction of clusters depended on the expression level of the Par-3, the expression levels of the full length and deltaCR1 constructs should be compared.

3) The authors say that cell membrane curvature was higher where Par-islands were attached. However, the plasma membrane is not shown. Thus, it is unclear what the curved, Par-3 detections represent (detections that were not obvious in all images). Imaging of a membrane marker in combination with Par-3 should be attempted to assess the membrane curvature directly.

4) The use of Par-6-GFP to monitor complexes induced by Par-3 over-expression is a concern. The Par-6-GFP clusters may be associated with Par-3 or may be a separate complex promoted by Par-3 (e.g. Par-6-aPKC in complex with Cdc42).

Please note that a concern was raised regarding the novelty of work, with much of data confirming the existing knowledge in the in vivo system. Thus, the authors are encouraged to emphasize the novelty of the work (re-constitution of asymmetry in non-polar cells) by textual editing, and/or by addressing the points above, which may be sufficient to address this concern.

Essential revisions that can be addressed by textual modifications:

1) The author showed that ROCK inhibitor, Y27632, did not significantly affect the behavior of Par-islands (subsection “Roles of Par components and the cytoskeleton in polarity formation”). Does it mean that the Par-islands dynamics is independent of Myosin cortical flow that requires ROCK function? If this is the case, will the dynamics of Par-islands require other actin-based motor proteins or not? I would like to see a discussion on this result.

2) Lgl and Par3 distribution was observed as opposite "crescents" in induced polarized S2 cells (Figure 2B). As far as I know, Lgl localization is uniformly cortical in *Drosophila* mitotic neuroblasts. It will be nice to include a short discussion/explanation on this different distribution of Lgl in these two cell types. In Figure 2B, ">UAS-Baz" was labelled in the figure. Should it be UAS-Par-3?

3) The authors should acknowledge the observation of Benton and St Johnston of spherical CR1 aggregates in the cytosol of oocytes (PMID: 12906794; Figure S1).

4) The authors say that Par-3 islands with amorphous and dynamic behavior have not been observed in *Drosophila* or *C. elegans*, but such organization seems similar to the clusters of Par-3 described in the one-cell *C. elegans* embryo or during initial epithelial polarization in *Drosophila*.

---

## [Author Response]

Essential revisions that may require additional experiments:1) The structures of the clusters formed with Par-3 or Par-3 lacking CR1 (the oligomerization domain) were not clearly different in the images shown. Perhaps the authors' super-resolution approach would reveal a clearer difference. This further assessment would be important for assessing whether the oligomerization domain has a role in forming filaments in cells. Depleting endogenous Par-3 may be needed for seeing a clear role of CR1.

We have shared the reviewers’ concern about the phenotype of the overexpression of Par3 lacking the CR1 oligomerization domain. If the CR1 was a sole domain that were involved in Par3-Par3 interaction, the lack of this domain should not generate Par3-oligomeric form or large structures such as the Par-islands. Indeed, there is a big difference between the aggregation state of Par complex after the overexpression of wild type Par3 and that of the Par ΔCR as described in the previous manuscript; the steady state S2 cells overexpressing wild type Par3 formed large islands, some of which were larger in area than 2 μm^2^ (the previous Figure 4A and the histograms in the new Figure 12). On the contrary, the overexpressed Par3ΔCR1 came to the cell cortex of S2 cells, and no longer form large islands. However, bright spots with a relatively small size and ambiguous contour were formed on the background of high level of Par complex distribution over the cortex (the previous Figure 10B). The question is why some aggregation structures were formed in spite of the lack of the known aggregation domain. To pursue this problem, we performed the combination of 3 new experiments; (1) elimination of the endogenous Par3 by RNAi, (2) observations by STED super resolution microscopy, and (3) the quantitative comparison in size between Par3-islands and Par3ΔCR1 bright spots at the steady state. Firstly, as suggested by the review, we tried to eliminate the endogenous Par3 by RNAi, and successfully reduced it to approximately 10% of the wild type S2 cells (Figure supplement 2). As the next step, we needed to introduce silent mutations into the RNAi sequence part of the wild type and Par3ΔCR1 in the overexpression construct so that the exogenous full-length Par3 and Par3ΔCR1 were not affected by RNAi. We confirmed by gel electrophoresis that the introduction of silent mutations into Par3 and Par3ΔCR1 did not significantly affect their expression level, when they were overexpressed under the same condition (Figure supplement 3). As the conclusion, the quantitative measurement indicated no significant contribution of the endogenous Par3 to the formation of these bright spots formed by the overexpression of Par3ΔCR1 (see below).

2) Under the RNAi condition, we examined the S2 cells overexpressing Par3-GFP and those expressing Par3ΔCR1-GFP in a STED microscope to obtain a super resolution images, as also suggested by the reviewers. We confirmed a various size of Par-islands including large ones with a regular meshwork structure when full length Par3-GFP was overexpressed (as we showed in the previous Figure 9). On the contrary, in the cells overexpressing Par3ΔCR1-GFP, we often observed bright spots with an amorphous shape, and could rarely find clear island structures with a regular meshwork structure. Thus, our STED observations strongly suggest that in the absence of Par3 CR1 domain, Par-islands consisting a meshwork with unit-like segments were not formed (see the new Figure 12).

3) To analyze how these bright spots formed by Par3ΔCR1 overexpression are different or similar to the islands formed by wild type Par3 overexpression, we measured the size of islands in S2 cells expressing Par3 (with silent mutations) and that of bright regions in S2 cells expressing Par3ΔCR1 (with silent mutations) after RNAi suppression of the endogenous Par3 at the steady state, and compared their distribution. We also compared the distribution of spot size with and without RNAi. We found that size distribution of bright spots did not significantly change upon RNAi suppression of the endogenous Par3, suggesting that those spots are not formed by the presence of the endogenous Par3 (data not shown). The results also show that the distribution of spot size after Par3ΔCR1 overexpression was quite different from that of islands formed by the overexpression of Par3; approximately 90% of bright spots are smaller in area than 1 μm^2^ (the mean value ± s.d. = 0.44 ± 0.37 μm^2^). These observations raise the possibility that Par3ΔCR1 still has an ability to form some cortical aggregates either via (1) an un-identified domain in Par3 or (2) via Par6-aPKC, or (3) via the other components. However, these aggregates do not form the regular meshwork as does the wild type Par3, and the majority of Par3ΔCR1-GFP distributes broadly in the cell cortex without a particular structure as revealed by STED imaging.

All together, we conclude that the CR1 domain of Par3 is critical for Par complex to form Par-islands with the regular meshwork and to asymmetrically localize in S2 cells. This conclusion is consistent with the previous finding that CR1 domain is an essential domain of the Par3 function to form the normal asymmetry in the cell cortex in the epithelium (Benton and St Johnston, 2003).

We have added these data in the new Figure 12 and Figure12—figure supplement 2 and Figure 12—figure supplement 3, and described these new results in a concise form in the text (subsection “Structural analysis of the assembly state of the Par-complex”) in the revised manuscript).

2) Since the induction of clusters depended on the expression level of the Par-3, the expression levels of the full length and deltaCR1 constructs should be compared.

Thanks a lot for a comment regarding a prerequisite for the comparison of the effect of Par3 overexpression with Par3ΔCR1 overexpression. We compared the expression levels of the full length Par3 and that of Par3ΔCR1 under the same condition used for the experiments (8 h after induction by metallothionein promoter). We performed Western blotting after gel electrophoresis of cells transfected by the expression plasmids for those proteins. As shown in Figure supplement 3. the result indicates that the expression level of the full length Par3 and Par3ΔCR1 is nearly at the same level. We, therefore, described this result in the text as follows; we measured the expression level of the full length Par3 and Par3ΔCR1 by Western blotting of cells transfected by the expression plasmids for those proteins, and confirmed that their expression levels are nearly at the same level (Figure12—figure supplement2).

3) The authors say that cell membrane curvature was higher where Par-islands were attached. However, the plasma membrane is not shown. Thus, it is unclear what the curved, Par-3 detections represent (detections that were not obvious in all images). Imaging of a membrane marker in combination with Par-3 should be attempted to assess the membrane curvature directly.

We are sorry for the lack of the image that simultaneously shows the plasma membrane and Par3 distribution and also the quantitative comparison of the curvature between the Par3 islands and the plasma membrane curvature. We have added these data as the new Figure 9 and described how the curvature of the islands is different from that of the plasma membrane after the first sentence of the subsection, ‘Structural analysis of the assembly state of the Par-complex”.

We expressed memGFP by the *actin*-promoter to visualize the cell membrane, together with myc-Par3-mKate2 that was induced by the *metallothionein*-promoter. The images for quantification were taken at eight hour after induction. The curvature radius for individual islands was obtained from three different points on the plasma membrane of individual islands. On the other hand, the curvature radius of the plasma membrane was calculated from the three different points along the cell membrane of less than 5 μm long. For this measurement, we chose the ‘non-neighbor region’, which was defined as the region along the cell membrane more than 5 μm away from the edge of neighboring Par-islands. This is to avoid a potential influence of Par-islands on neighbouring membrane curvature (please see Materials and methods section for technical detail). As shown in the new Figure 9, the curvature radii of Par-islands were in the range 0.7 μm to 5 μm with the median of 1.70 μm. (n = 69 islands in 34 cells). On the contrary, the majority of curvature radii for the plasma membrane was in the range between 2 μm and 20 μm with the median of 4.07 μm in the non-neighboring regions (n = 87 plasma membrane regions in 29 cells). We also measured the curvature radii of non-Par-islands cells lacking Par-island formation, which gave us the median of 4.56 μm with the range of 2 μm and 20 μm (n = 76 regions in 13 cells). These distributions of the curvature radii of the three different regions indicate that the curvature radius of Par-islands is significantly smaller than that of cell membrane curvature in non-neighboring regions, which is virtually consistent with the radius of S2 cells that we used for experiments. Accordingly, Par-islands show a more convex shape than the ordinary plasma membrane curve of S2 cells.

4) The use of Par-6-GFP to monitor complexes induced by Par-3 over-expression is a concern. The Par-6-GFP clusters may be associated with Par-3 or may be a separate complex promoted by Par-3 (e.g. Par-6-aPKC in complex with Cdc42).

Thank you for pointing out a key point of our experimental design.

We initially checked whether the distributions of Par3-mKate2 and Par6-GFP are identical to each other, when we used the metallothionein induction system as a promoter to monitor the temporal pattern of Par complex aggregate formation. We needed a fluorescent marker, intensity of which was kept constantly during induction until reaching the steady state, because at the early stage of of induction of Par3-mKate2, mKate2 fluorescence was too weak to detect. Par6-GFP was a good marker, because its intensity is kept nearly constant during Par3-mKate2 induction, as shown in Discussion Figure supplement 1 in the initial manuscript.

We, therefore, compared the cortical expression pattern of Par3-mKate2 and Par6-GFP at 4 hours in S2 cells after induction of Par3-mKate2. We found that the spatial expression pattern of both fluorescences almost overlapped at the resolution of the confocal microscopy while some differences in the local intensity gradient between Par6-GFP and Par3-mKate2 (this may partly come from the difference in the setting of fluorescence detection for each wavelength). There were also a few small dots that appears to contain only either fluorescence in one cell image.

We understand reviewers’ concern because Par3-Par6-aPKC complex and cdc42-Par6-aPKC complex are exclusively formed, playing different roles (for example, Rodoriguez et al., 2017). During revision of the manuscript, we therefore, compared the distribution of Par3-mKate2, Par6-GFP and Cdc42 4 h after Par3-mKate2 induction by immunostaining of fixed samples. At 3 hours post induction, small Par-islands have been formed. As described above, the distribution of Par3-mKate2 and Par6-GFP are almost identical with each other. In contrast, Cdc42 distribute quite differently from these of Par components. Cdc42 distributes in a dotty or short string patterns (Slaughter et al., 2013, and Sartrel et al., 2018). Essentially, the pattern of Cdc42 distribution has no correlation with those of Par3-mKate2 and Par6-GFP, while some Cdc42 dots appear to be located at the edge of Par-islands. At 8 hours, we have confirmed the nearly identical spatial distribution of Par3-mKate2 and Par6-GFP during observations of many cells expressing both Par3-mKate2 and Par6-GFP for the SRRF analysis (the previous Figure 9 → new Figure 11). We have insert several sentences at the beginning of the subsection “Temporal patterns of Par-complex polarization”,to describe the rationality for the monitoring Par-complex temporal pattern with Par6-GFP during induction and at the steady state is based on the observations of (1) their almost identical distribution at 3 h after induction and (2) not apparent relationship of their relationship with that of Cdc42. We have also added these data as the new Figure 4.

*Please note that a concern was raised regarding the novelty of work, with much of data confirming the existing knowledge in the* in vivo system. Thus, the authors are encouraged to emphasize the novelty of the work (re-constitution of asymmetry in non-polar cells) by textual editing, and/or by addressing the points above, which may be sufficient to address this concern.Essential revisions that can be addressed by textual modifications:1) The author showed that ROCK inhibitor, Y27632, did not significantly affect the behavior of Par-islands (subsection “Roles of Par components and the cytoskeleton in polarity formation”). Does it mean that the Par-islands dynamics is independent of Myosin cortical flow that requires ROCK function? If this is the case, will the dynamics of Par-islands require other actin-based motor proteins or not? I would like to see a discussion on this result.

We have reproducibly obtained the data showing no significant effect of Y27632 on the behavior of Par-islands. Because Myosin II is essential for *C. elegans* cortical flow, this observation suggests that the dynamic behavior of Par-islands is not dependent on the typical Myosin II cortical flow. As we described in the Discussion section in the previous manuscript, we have also observed no directional movement of Par-dots or Par-islands during the process of asymmetric clustering and at the steady state. Based on these two observations, we described in the previous manuscript that we failed to obtain evidence showing an involvement of cortical flow in Par-island dynamics (subsection “Temporal patterns of Par-complex aggregation”). As far as we checked literature, Y27632 compromises the function of all Myosins bearing MLCs either or both via inhibiting MLCK and activating MLC phosphatase. However, if there is a myosin isotype insensitive to Y27632, we do not exclude the possibility that such Y27632-insensitive Myosin might work for the dynamics of Par-islands even though it is unlikely to operate in a directional way. In summary, we have currently no positive evidence that Myosin and cortical flow are involved in the dynamics of Par-islands.

How can Par islands cluster asymmetrically and dynamically behave? We can argue another possible mechanism underlying Par-island dynamics, which is diffusion and/or tension. We think that the situation where advection such as cortical flow surpasses diffusion for transportation is the following two cases. Firstly, the scale of the system is large enough. *C. elegans* one cell embryo and *Drosophila* oocytes are good examples. In these cases, the random diffusion and capture mechanism without advection will take a much longer time (Goehring et al., 2011). While *C. elegans* one cell embryo is approximately 50 μm long along the antero-posterior axis, S2 cells are approximately 10 μm in diameter, which seems to be nearly marginal about which of advection and diffusion is efficient to polarize cells. The mutual capture of islands might be sufficient to cluster islands in one pole.

Secondly, the time scale of clustering in S2 cell system is an order of hours, while the polarization of *C. elegans* one cell embryo or of *Drosophila* neuroblasts occurs in an order of minutes (less than 5 min for *Drosophila* neuroblasts). Time scale of polarization of Par-islands, together with its non-directional movement, is likely explained by the notion that polarized clustering of Par-islands occurs through diffusion and mutual capture. Mutual capture of Par-islands might be driven by phase separation with surrounding cortex.

Measurement of the trajectory of Par-island movement and plotting the mean square distance will be the best way to test whether the movement depends on diffusion or advection, but frequent mutual fusion and fission of islands makes it technically difficult to accurately trace the trajectory. This will be an important future issue while it is beyond scope of this paper. In conclusion, while we do not exclude the involvement of myosin motor in Par-island dynamics, it is more likely to be driven by diffusion mechanism on the two-dimensional cortex.

We have made a section, ‘Mechanism of clustering of Par-islands,’ and discussed this issue in a more concise way.

2) Lgl and Par3 distribution was observed as opposite "crescents" in induced polarized S2 cells (Figure 2B). As far as I know, Lgl localization is uniformly cortical in *Drosophila* mitotic neuroblasts. It will be nice to include a short discussion/explanation on this different distribution of Lgl in these two cell types.

We agree that in the first manuscript the differences between our S2 cell polarity system and *Drosophila* neuroblasts were not sufficiently explained. Essentially, Lgl relocalizes to the cytoplasm during mitosis while it is cortically retained during interphase.

According to the previous literature (Ohshiro et al., 2000, and Pen et al.,2000), Lgl uniformly distributes along the cell cortex of *Drosophila* embryonic neuroblasts ‘at interphase’, but once neuroblasts enter ‘mitotic phase’, Lgl quickly drops off the cortex into the cytoplasm. Later on, it was shown that Lgl becomes complementary to the distribution of the apical complex during early prophase, and shortly disappears from the entire cell cortex via the function of Aurora A kinase by the end of prophase (Wirtz-Peitz, et al., 2008). This was shown by using of *Drosophila* sensory precursor cells and larval neuroblasts. More recently, Lgl relocalizes to the cytoplasm via phosphorylation of two of among the three functional phosphorylation sites on Lgl by AurA and B in *Drosophila* epithelial cells (Bell et al., 2015). aPKC phosphorylation site is the third one, which operates on polarized distribution of the cortically localizing Lgl.

Because AurA and B become active only during mitotic phase in ordinary cells, Lgl normally localizes at the cell cortex during interphase. This is also the case for S2 cells and *Drosophila* neuroblasts. In this study, we overexpressed Par3 in S2 cells, majority of which is at interphase (approximately 24 hour cell cycle length), and could have polarized them upon Par3 overexpression at interphase. Therefore, the complementary cortical distribution of the Par complex and Lgl was stably maintained during interphase. This situation is different from the case of neuroblasts where Par complex is activated by AurA upon mitotic entry.

We have mentioned to Lgl distribution briefly in the Discussion part describing the differences in cell polarization between in our S2 cell system and in *Drosophila* neuroblasts in the Discussion section.

In Figure 2B, ">UAS-Baz" was labelled in the figure. Should it be UAS-Par-3?

Thank a lot for pointing out the nomenclature of Par3 (Baz in *Drosophila*). We decided to Par3 for easy understanding for readers in non-*Drosophila* fields.

3) The authors should acknowledge the observation of Benton and St Johnston of spherical CR1 aggregates in the cytosol of oocytes (PMID: 12906794; Figure S1).

We are very sorry about the failure of citation of important references when we described the cytoplasmic aggregates of Par3 mutant missing membrane binding domain. Whereas we cited this paper itself as one of the first two papers that described that the N-terminal part of Par3 (CR1) has an ability of oligomerization (the third reference in the previous manuscript), we failed to cite it in a complete form due to the trouble of a reference software we used. We also failed to cite this paper once more when we described the behavior of the mutant form lacking the C terminal domain in the cytoplasm.

We also have several papers that we should have referred to as several papers as described in the response to comment 4.

4) The authors say that Par-3 islands with amorphous and dynamic behavior have not been observed in *Drosophila* or *C. elegans*, but such organization seems similar to the clusters of Par-3 described in the one-cell *C. elegans* embryo or during initial epithelial polarization in *Drosophila*.

We do not agree the reviewer’s view about ‘Par-3 islands with amorphous and dynamic behavior’ as [Par-3 islands with amorphous and dynamic behavior seems similar to the cluster of Par3described in one-cell *C. elegans* embryo or during initial epithelial polarization in *Drosophila*].

We think that the question is at what resolution, in which system, researchers want to examine Par complex aggregates and their behavior. In other words, how the word ‘similar’ is defined. Below are our responses.

1) *Drosophila* neuroblasts, we are confident that any previous work has not described the structure of the apical crescent in detail about whether the shape of Par-complex clusters are amorphous or robust, or how dynamically the fusion and fission of clusters take place.

2) *C. elegans* one cell embryo; there are so many studies on the Par complex clusters and their behavior, some of studies classified Par-complex cluster in size (and bright) as described in the DevCell paper from Goldstein lab (Dickinson et al., 2017). However, the shape of the large Par-cluster was not examined in this paper. As for the behavior, they analyzed extensively and separated the advective component and random (maybe Brownian) component. Their analysis shows that the advective flow component is major for large bright spots, which might correspond to our Par-islands regarding the size of particles, while we could not separate the movement and association/dissociation of Par-islands. On the other hand, a random behavior is dominant in dim spots. In addition, there is no description about association and dissociation behavior. High resolution image analysis is not included in this paper because of a difference of scope. There is also no description about internal structures. So, we cannot judge whether these clusters are similar or not. Another paper was also published in 2017 from Fumio Motegi lab in Singapore (Wang et al., 2017). In this paper, there is a lot of movies as supplemented data. Some of them, for example, video 13 includes Par-complex particles showing a behavior similar to the Par-islands. There was no description about it in the manuscript, however.

3) Initial epitheialization in *Drosophila*. It has been known that there are patches of Baz and other adhesion-related molecules in early epithelial polarization, mainly by contributions of Tony Harris and Mark Peifer. We are not sure about whether there is a detailed description of shape and behavior with fusion and fission, although we are familiar to these papers. So we decided to describe and cite two papers (Harris and Peifer 2005, Harris and Peifer, 2007). It might be interesting to examine the fine structure of those patches by super-resolution microscopy to see whether meshwork structures exist or not.

4) *Drosophila* female germline stem cells. We were not aware of the paper describing the Baz/Par3 patches. which is small in size during interphase (Inaba et al., 2015), whereas we have noticed large Baz/Par3 and Cadherin region in female germline stem cells. While there is not description about dynamism or association or dissociation during expansion, it would be interesting to see it.

Although we do not think that so far there is not detail description regarding dynamics, fission and fusion, and the fine structure about ‘similar clusters’, we have weakened the expression of [To our knowledge, this structure has not been reported in cortical Par-complex assembly in *C. elegans* or *Drosophila*]. “*C. elegans* or *Drosophila*’ has been changed to ‘*C. elegans* one cell embryo and *Drosophila* neuroblasts’, and we referred to as the papers described above.